# EXPLORATORY CAUSAL INFERENCE IN SAENCE

**Tommaso Mencattini**[†,1,2]**, Riccardo Cadei**[†,1]**, Francesco Locatello**[1]

[1]Institute of Science and Technology, Austria (ISTA)
[2]École Polytechnique Fédérale de Lausanne (EPFL)
[†]*Equal contribution.*

## ABSTRACT

Randomized Controlled Trials are one of the pillars of science; nevertheless, they rely on hand-crafted hypotheses and expensive analysis. Such constraints prevent causal effect estimation at scale, potentially anchoring on popular yet incomplete hypotheses. We propose to discover the unknown effects of a treatment directly from data. For this, we turn unstructured data from a trial into meaningful representations via pretrained foundation models and interpret them via a sparse autoencoder. However, discovering significant causal effects at the neural level is not trivial due to multiple-testing issues and effects entanglement. To address these challenges, we introduce *Neural Effect Search*, a novel recursive procedure solving both issues by progressive stratification. After assessing the robustness of our algorithm on semi-synthetic experiments, we showcase, in the context of experimental ecology, the first successful unsupervised causal effect identification on a real-world scientific trial.

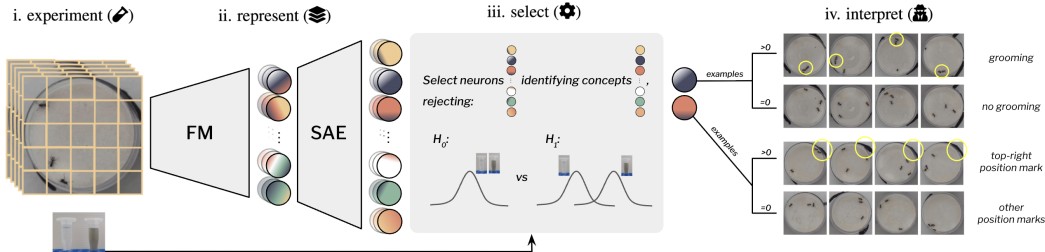

Figure 1: Pipeline for Exploratory Causal Inference: (i) collect experimental data, (ii) extract interpretable representations via a *Foundation Model* and *Sparse Autoencoder*, (iii) identify treatment effects, e.g., via *Neural Effect Search*, and (iv) interpret the causal findings, i.e., affected neurons.

## 1 INTRODUCTION

In science, data has been historically collected to answer specific questions (Popper, 2005). In this *rationalist* view, scientists formulate a hypothesis, often as a causal query, and collect data to falsify it. For example, experimental ecologists studying social immunity in ants (Cremer et al., 2007) may suspect that exposure to a specific micro-particle may affect specific social behaviors, or more formally, "a *treatment* $T$ has a causal effect on an *outcome* behavior $Y$". They then perform a controlled experiment, administering the treatment or a placebo to a sample of individuals and check whether there is a significant difference in the correlation between the treatment assignment and the outcome. While this paradigm has dominated science for centuries, modern science has started embracing the creation of *atlases*: vast, comprehensive maps of natural phenomena, collected for general purposes. Today, we have planetary-scale maps of life genomes (Chikhi et al., 2024), sequencing of 33 different types of cancer (Weinstein et al., 2013), and imaging of cells under thousands of perturbations (Sypetkowski et al., 2023) to name a few. Different than the classical paradigm, these datasets call for an *empiricist* view, starting with exploratory data-driven investigations. The new challenge is that the immense size of these datasets prohibits scientists from simply "looking at the data and finding out what is interesting". Even beyond atlases, in

our motivating example from experimental ecology, fine-grained social interactions between many individuals are complex yet critical for understanding the spread of a disease (Finn et al., 2019). Hopefully, data processing can be dramatically accelerated with modern machine learning, e.g., relying on pre-trained foundational models and leveraging artificial predictions within causal inference pipelines (Cadei et al., 2025). Still, scientists need to decide what to "look at" a-priori before to validly annotate the data and infer causal relationships, potentially biasing the analyses towards recurrently studied behaviors, phenomena known as the "Matthew effect" (Merton, 1968) or informally as "rich-get-richer": scientists are biased by prior successful investigations.

---

**Motivating Example:** *behavioral effects in Social Immunity research*

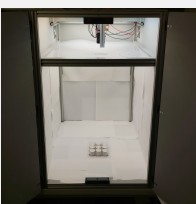

*Experimental ecologists collect video data of ants' interactions under different treatments to study social immunity policies manifested in specific behaviors (Cremer et al., 2007; Cadei et al., 2024). In practice, these behaviors are not directly measured, but annotated from the videos by hand or using machine learning. Which behaviors to study is determined by the **scientists' prior** (not data driven), which may **not** be a priori **clear or exhaustive in exploratory experiments**.*

---

In this paper, we characterize differences and synergies between the classical *rationalist* view and the emerging *empiricist* one and propose a method to identify statistically significant effects in exploratory experiments, formally grounding it with the language of statistical causality, see Figure 1. We formulate the problem on experimental data, where a treatment is administered randomly and the possible effects are measured indirectly, e.g., via imaging or other raw observations. Instead of requiring scientists to formulate treatment effect hypotheses, label some data, and train a model to extend labels to the whole dataset (i.e., the rationalist view (Cadei et al., 2024; 2025)), we propose to train a sparse autoencoder (SAE) on the representation of a foundation model and generate data-driven effect hypotheses interpreting the significant effects on the neural representations (i.e., empiricist approach). In this new paradigm, the main challenge is that, if the SAE is not *perfectly* disentangled (Elhage et al., 2022), any neuron minimally entangled with the true effect may appear significantly treatment-responsive, challenging the results interpretation. To address it, we propose a novel recursive stratification technique to iteratively correct the effect on entangled neurons.

Looking at the data before committing to any hypothesis, we overcome the Matthew effect, enriching the rationalist view in a data-driven way. We propose to work with pretrained foundation models, training SAEs directly on the target experimental data. This is important because pretrained foundation models can be biased as well, which is problematic for drawing scientific conclusions (Cadei et al., 2024). Instead of directly testing a single hypothesis, our approach enables to preliminary explore thousands of potential effects in a semantically expressive latent space, still allowing the domain experts to interpret, judge and eventually test them a posteriori. This is in stark contrast with preliminary empiricist approaches in causality like "causal feature learning" (Chalupka et al., 2017), which only commits to a single hypothesis by discrete clustering. Our contributions are:

- Within the statistical causality framework, we formally **differentiate rationalist and empiricist approaches** to causal inference, highlighting their complementary strengths and limitations.

- We propose a **novel empiricist methodology** building on foundation models and sparse autoencoders. We characterize the statistical challenges in multiple hypothesis testing to discover treatment effects over neural representations in the *paradox of exploratory causal inference* (ours). Then, we introduce *Neural Effect Search*, a novel iterative hypothesis testing procedure to overcome such challenges.

- We showcase in both semi-synthetic (real images with controlled causal relations) and a real-world trial in experimental ecology that our approach is capable of identifying the treatment effects in a exploratory experiment. To the best of our knowledge, this is the **first successful application** of sparse autoencoders to causal inference, tailored to the analysis of scientific experiments.

## 2  TREATMENT EFFECT ESTIMATION IN RANDOMIZED CONTROLLED TRIALS

**Notation.** *In the paper, we refer to random variables as capital letters and their realizations as lowercase letters. Matrices are referred to as upper-case, boldface letters.*

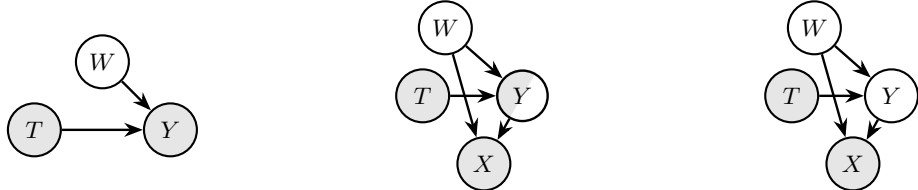

Figure 2: Exemplary graphical models for randomized controlled trials to infer the treatment effect (in light gray the observed variables). In classic **Causal Inference (left)**, both the treatment $T$ and outcome of interest $Y$ are observed and unconfounded. In **Prediction-Powered Causal Inference (center)**, $Y$ is only measured indirectly via a complex high-dimensional measurements $X$, partially labeled. The missing $Y$ are retrieved from $X$, training a predictive model on the annotated data. In **Exploratory Causal Inference (right)**, the affected outcome $Y$ is unknown, still measured within $X$, from which a model aims to identify it.

**Causal Inference.** Causal Inference aims to quantify the effects of an intervention on a certain variable *treatment* on some *outcome* variables of interest, see Figure 2 (left). For simplicity, we consider a binary treatment $T = \{0, 1\}$ (e.g., taking a placebo or a drug) and outcome variables $Y \in \{0, 1\}^r$ (e.g., binary indicators for symptoms, biomarkers, or clinical events). While continuous extensions would be interesting, we focus on discrete outcomes since continuous concepts in SAEs are not well understood yet (Quirke et al., 2025). At population level we aim to estimate the *Average Treatment Effect* (ATE):

$$\tau = \mathbb{E}[Y(T = 1) - Y(T = 0)], \tag{1}$$

where $Y(T = 1)$ and $Y(T = 0)$, or $Y(1)$ and $Y(1)$ for short, denote the potential outcomes under treatment and control (Rubin, 1974) (equivalently $Y|do(T) = 1$ and $Y|do(T) = 0$ according to Pearl (2009)). Estimating $\tau$ is challenging because, for each individual, only one potential outcome is factually observed—the one under the received treatment—so the counterfactual is missing (fundamental problem of causal inference (Holland, 1986)). This problem is mitigated in the sciences by performing, whenever possible, a *Randomized Controlled Trial* (RCT). By randomly assigning the treatment (i.e., $T$ has no causes), we prevent spurious correlations between the treatment and any other cause $W \in \mathbb{R}^q$ of the outcome (no confounders), allowing to (statistically) identify the ATE with the associational difference, i.e.,

$$\tau = \mathbb{E}[Y \mid T = 1] - \mathbb{E}[Y \mid T = 0], \tag{2}$$

under standard causal assumptions (Rubin, 1986) of consistency (observing $T = t$, then $Y = Y(t)$), and no interference across individuals (i.e., all individuals are independent samples from the population, and the treatment assignment to the individual $i$ does not affect individual $j$). It follows that the difference between the treated and control groups' sample means is already an unbiased estimator of the ATE. Nonetheless, more sophisticated estimators such as Augmented Inverse Propensity Weighting (AIPW (Robins et al., 1994)) can achieve lower variance and thus greater efficiency.

**Prediction-Powered Causal Inference and the *rationalist approach*.** Assume that $Y$ is latent and not observed directly. Instead, we observe it indirectly within high-dimensional measurements $X \in \mathcal{X} \subseteq \mathbb{R}^p$, capturing all the affected outcome information, i.e., $H(Y|X) = 0$, mixed with the other (latent) outcome causes $W \in \mathbb{R}^q$. In our motivating example, let $Y$ representing a well-defined behavior of interest and $X$ its video recording, for which we don't have a perfect behavior classifier yet. Prior work by Cadei et al. (2024; 2025) showed how to train a model on partially labeled data or similar experiments to predict factual outcomes $\hat{Y} \approx Y$ from $X$ and then use them for causal inference. For simplicity, we assume that $T$ is not directly observable in $X$, a common practice in double-blind randomized trials (e.g., neither the patient nor the doctor analyzing the results knows which treatment was assigned). The set-up is illustrated in Figure 2 (center). Both Causal Inference and Prediction-Powered Causal Inference represent *rationalist* approaches in scientific discovery, ignoring the hypothesis generation preliminary, i.e., data-driven $Y$ definition.

**Exploratory Causal Inference and the *empiricist approach*.** The rationalist approaches suffer the Matthew effect (Merton, 1968), narrowing the effect hypotheses on the outcome of prior successful trials. In this paper, we consider the setting where experiments are *exploratory*, which we informally model as the scientists having no or partial *a priori* knowledge of what the latent effect may be. This is shown in Figure 2 (right), with $Y$ representing the unobserved and unknown latent factors affected by the treatment T and measure in $X$. In principle, one may consider the pixels themselves as influenced by the treatment. We instead consider the ground truth $Y$ to be the *coarsest* possible abstraction of the treatment effect in $X$ (Rubenstein et al., 2017; Chalupka et al., 2017). In other words, $[Y, W]$ is the minimal and sufficient representation for $X$ (Achille and Soatto, 2018; Fumero et al., 2023) that renders the treatment independent from the observations, i.e., $T \perp\!\!\!\perp X | [W, Y]$, where only $Y$ is causally affected by a controlled intervention on $T$. With a slight abuse of notation, we do not need to assume that such $Y$ exists, so $r$ can be zero if the treatment has no effect at all. Our goal is to propose candidate effects $Y$ to the scientists in a purely data-driven way, discovering significant statistics that differentiate the treated and control populations. It is important to remark that we do not interpret these statistics as necessarily scientifically relevant. The reason is that, when working with high-dimensional data, there can be irrelevant effects, i.e., visible treatment and (finite sample) experiment design biases. Our approach is to identify *all* significant statistics and leave the interpretation to the domain experts. The empiricist view should not replace the rationalist one, but enrich it with additional data-driven hypotheses.

## 3 EXPLORATORY CAUSAL INFERENCE VIA NEURAL REPRESENTATIONS

To detect treatment effects when only high–dimensional indirect outcome measurements $X$ are available, we turn these raw observations into analyzable measurements. We first pass samples $x$ through a pretrained foundation model (FM) (Bommasani et al., 2022), obtaining representations $h = \phi(x) \in \mathbb{R}^d$ whose geometry captures semantically meaningful regularities (Amir et al., 2022; Valeriani et al., 2023). Throughout, we assume the FM is *sufficient for the outcome information* (Achille and Soatto, 2018), i.e., $I(X, Y) = I(\phi(X), Y)$, so working in $h$ preserves exactly the information about the (unknown) outcome factors $Y$ that is present in the raw data. Under sufficiency, any arm difference that exists in $X$ is detectable in representation space, making $h$ a principled surrogate for measurement.

**From FM features to a measurement dictionary.** While FM features are semantically structured, individual coordinates in $h$ generally not align with human–readable concepts (Bricken et al., 2023). We therefore *reparameterize* the representation into a sparse, interpretable measurement dictionary using a sparse autoencoder (SAE) (Bricken et al., 2023; Huben et al., 2024). Intuitively, the SAE expresses each $h$ as a sparse linear combination of atoms that behave like measurable channels; sparsity biases solutions toward localized, approximately monosemantic features that scientists can inspect a posteriori. Given foundation model's features $h \in \mathbb{R}^d$, the SAE computes a high–dimensional but *sparse* code $z \in \mathbb{R}^d$ and reconstructs $h$ linearly:

$$z = f(h) = g\left(\mathbf{E}^\top h + b_e\right), \qquad \hat{h} = \mathbf{D}z + b_d, \tag{3}$$

where $\mathbf{E}, \mathbf{D} \in \mathbb{R}^{d \times m}$ are respectively the encoder, and decoder linear maps, $b_e, b_d \in \mathbb{R}^m$ are the learnable biases, and $g : \mathbb{R}^m \to \mathbb{R}^m$ is the encoder nonlinearity (Bricken et al., 2023). Training minimizes a reconstruction loss with a sparsity penalty $\mathcal{S}$ weighted $\lambda \geq 0$, i.e.,

$$\min_{D, z \geq 0} \mathbb{E}\left[\|h - \mathbf{D}z - b_d\|_2^2\right] + \lambda \mathcal{S}(z). \tag{4}$$

Thereafter, each input can be summarized as $h \approx b_d + \sum_j z_j d_j$, where $\|z\|_0 \ll d$ and $\mathbf{D} = [d_1, \ldots, d_m]$. This turns the FM representation into a large dictionary of interpretable channels: each coordinate $z_j$ serves as a putative detector of a simple attribute, with still some inevitable leakage (Huben et al., 2024).

**Monosemanticity, leakage, and entanglement.** In exploratory experiments, we would like each SAE code coordinate to behave like a single, human–readable measurement channel for a simple outcome factor. When this happens, a scientist can read off "what changed" from the few activated codes. In practice, however, codes often *leak* across factors: one neuron can respond weakly to several distinct attributes, i.e., weak *polysemanticity* and corresponding *entanglement* (Locatello et al., 2019). We need a minimal language to talk about (i) the direction in code space associated

with a factor and (ii) how widely those directions spill across neurons. Let $Z \in \mathbb{R}^m$ be SAE codes and $Y = (Y_1, \ldots, Y_r)$ the (unknown) binary outcome factors with $m \gg r$. To define the leakage set and index, we first define the concept $Y_k$ neural representation as:

$$v_k := \mathbb{E}[\, Z \mid do(Y_k = 1)\,] - \mathbb{E}[\, Z \mid do(Y_k = 0)\,] \in \mathbb{R}^m \quad \forall k \in \{1, \ldots, r\}. \qquad (5)$$

and we say that the neuron $Z_j$ with $j \in \{1, \ldots, m\}$ is *activated* by the factor $Y_k$ if $|(v_k)_j| \geq \varepsilon > 0$. When the neural effect representations $\{v_k\}_{k=1}^r$ are *sparse* and largely *disjoint* across coordinates, each effect factor "lights up" only a few neurons, and different factors rely on different neurons.

**Definition 3.1** (Leakage set and index). *Fix a threshold $\varepsilon > 0$. In a ECI problem with effect neural representations $\{v_k\}_{k=1}^r$, we define the* leakage set *and* leakage index*, respectively, as*

$$\mathcal{A}_\varepsilon = \bigcup_{k=1}^r \{\, j : |(v_k)_j| \geq \varepsilon \,\}, \qquad \rho_\varepsilon := \frac{|\mathcal{A}_\varepsilon|}{m}. \qquad (6)$$

If $|\mathcal{A}_\varepsilon| \gg r$, i.e., $\rho_\varepsilon \gg 0$, it indicates that many neurons respond to multiple factors, i.e., polysemanticity, whereas monosemanticity with respect to $Y$ implies $|\mathcal{A}_\varepsilon| = r$, i.e., $\rho_\varepsilon \approx 0$.

**Codes as statistical measurement channels.** Under FM sufficiency and an (approximately) monosemantic SAE, it becomes natural to pose causal questions at the level of individual codes. If the true affected outcomes $Y$ are perfectly localized in disjoint subsets of coordinates of $Z$, then one can test each coordinate for a treatment–control mean shift using a two–sample $t$–test, applying Bonferroni adjustment (Bonferroni, 1936) to control the family–wise error rate at $\alpha$ regardless of the number of tests $m$. This provides an idealized measurement interface: we can scan $Z$ for treatment–responsive channels and later interpret significant coordinates via the dictionary atoms $d_j$.

**Challenge: entangled effect representations.** The above picture breaks down when leakage occurs, as any neuron entangled with the true affected outcome will eventually be identified as significantly activated, while $|\mathcal{A}_\varepsilon| = O(m) \gg r$, challenging any interpretation. Intuitively, entangled neurons that are primarily assigned to other concepts still activate differently depending on $Y$, so with more powerful tests (larger sample sizes or stronger causal effects), they would be deemed significantly affected. Thereafter, classical multiplicity correction does not rescue interpretability here, leading to the paradox of Exploratory Causal Inference:

> **Paradox of Exploratory Causal Inference**
>
> In Exploratory Causal Inference, as the trial sample size $n$ or the effect magnitude $\tau$ grows, multiple testing, even with Bonferroni adjustment, redundantly returns all the outcome-entangled neurons as independent and significantly affected by the treatment.

We formalize these two phenomena below. Let $\tau_j$ denote the treatment effect on code $j$.

**Theorem 3.1** (Significance level collapse with sample size). *Suppose at least $\rho_\varepsilon m$ neurons have nonzero effect $|\tau_j| \geq \varepsilon > 0$. Via multiple testing, regardless of the Bonferroni adjustment,*

$$\Pr\Big[\{\text{all } j \in \mathcal{A}_\varepsilon \text{ are rejected}\}\Big] \to 1 \quad \text{as } n \to \infty,$$

*and the number of rejections converges to $\rho_\varepsilon m$ in probability.*

*Proof sketch.* For each $j$, the $t$–statistic is asymptotically normal with noncentrality $\lambda_j = \sqrt{n}\,\tau_j/\sigma$. The Bonferroni cutoff, which determines the significance of $\tau_j$, grows like $\sqrt{2 \log m}$; this cutoff is dominated by the growth in expectation of $\tau_j$ ($\sqrt{n}$ as $n \to \infty$). Hence, any $j$ with $\tau_j \neq 0$ is eventually rejected. Without Bonferroni correction, the significance cutoff is constant. See full proof in Appendix A. □

**Theorem 3.2** (Significance collapse with effect magnitude). *Fix $n < \infty$ and let $\tau_j(s) = s\,\gamma_j$ with $s > 0$. Via multiple testing, regardless of the Bonferroni adjustment,*

$$\Pr\Big[\{\text{all } j \in \mathcal{A}_\varepsilon \text{ are rejected}\}\Big] \to 1 \quad \text{as } s \to \infty,$$

> *and the number of rejections converges to $\rho_\varepsilon m$ in probability.*

*Proof sketch.* The noncentrality $\lambda_j(s) = \sqrt{n}\, s\gamma_j/\sigma$ grows linearly in $s$, while the cutoff, even with Bonferroni correction, is fixed for fixed $m, n$; every $\gamma_j \neq 0$ is eventually rejected. Details in Appendix A. $\qquad\square$

**Numerical illustration.** Let $T \sim \text{Bernoulli}(0.5)$, $Y \mid T = t \sim \mathcal{N}(\tau t, 1)$ (single effect), and $Z = [Z_A, Z_B] \in \mathbb{R}^m$ where $Z_A = Y$ (the effect principal channel) and $Z_B \mid Y = y \sim \mathcal{N}(0.01\, y\, \mathbf{1}_{m-1},\, I_{m-1})$ (entangled channels). As shown in Figure 3 for 10 seeds, increasing either $n$ or $\tau$ leads all the multiple testing to flag all the weakly entangled $Z_B$ coordinates as significant, despite their negligible semantic relevance. This motivates the disentangling, stratified testing procedure introduced next (Section 4).

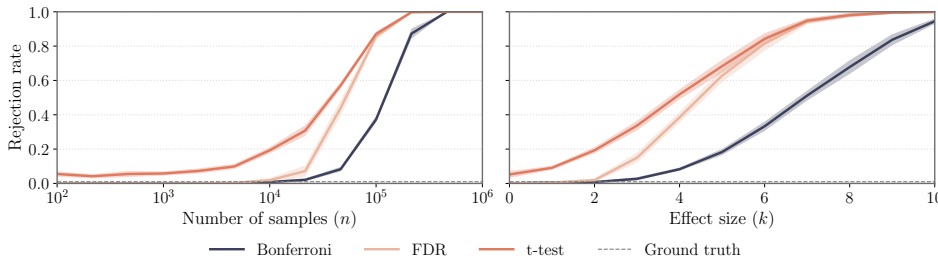

Figure 3: **The Paradox of Exploratory Causal Inference**: Increasing the power of the test, the effect on any outcome-entangled code becomes significant, regardless of its main interpretation.

## 4 NEURAL EFFECT SEARCH

To mitigate the multi-test significance collapse with entangled representation, i.e., Paradox of Exploratory Causal Inference, we propose NEURALEFFECTSEARCH (NES), a novel causally-principled algorithm to disentangle the leaked effects by recursive stratification:

---

**Algorithm 1** Neural Effect Search (NES)

---

1: **function** NEURALEFFECTSEARCH($T, Z, \alpha, \mathsf{S} = \varnothing$)
2:     $m \leftarrow \#\{\, j : j \notin \mathsf{S} \,\}$                         $\triangleright$ number of hypotheses to test
3:     **for** each neuron $j \notin \mathsf{S}$ **do**
4:         $(\hat{\tau}_j, p_j) \leftarrow$ NEURALEFFECTTEST($T, Z, j, \mathsf{S}$)           $\triangleright$ $p_j$ tests $H_0 : \tau_j = 0$
5:     **end for**
6:     $\mathsf{R} \leftarrow \{\, j \notin \mathsf{S} : p_j < \alpha/m \,\}$, ordered by $|\hat{\tau}_j|$ (desc)     $\triangleright$ filter significant neurons
7:     **if** $\mathsf{R} = \varnothing$ **then**
8:         **return** $\mathsf{S}$
9:     **else**
10:         **return** NEURALEFFECTSEARCH($T, Z, \alpha, \mathsf{S} \cup \mathsf{R}_1$)
11:     **end if**
12: **end function**

---

where NEURALEFFECTTEST (Algorithm 2) is the procedure for multi-hypothesis testing on all the neurons $j$, by stratification (and potentially arm-wise residualization) over the already retrieved effects $\mathsf{S}$. See Figure 4 for a simplified illustration of NES procedure, Appendix B for NEURALEFFECTTEST description and Appendix C for a minimal Python implementation snippet. The key idea is that if we test all neurons simultaneously, vanilla multiple testing cannot distinguish whether a neuron carries its own causal effect or merely leaks information about another concept. By contrast, NES first recovers the most prominent effect by its most representative neuron, then it *controls* its effect in subsequent tests, preventing the ECI paradox, continuing iteratively. Since this is a multiple testing setting, in Line 6 we still perform Bonferroni correction by dividing the significance level $\alpha$ by $m$, preventing I type error, i.e., false neural effects discoveries. In practice, if the sample size is

very small, one can make the test less conservative by relaxing the correction (which would return more, possibly false positives for the scientist to review).

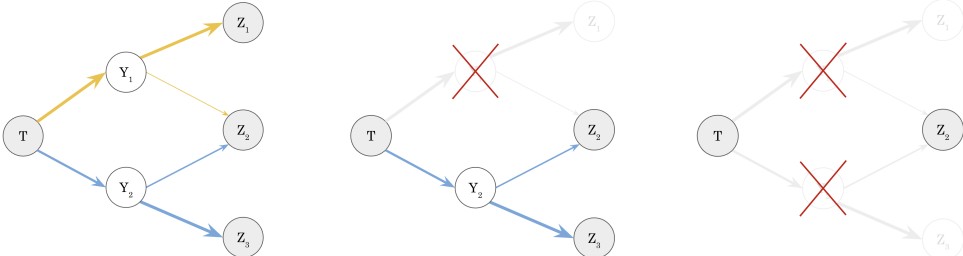

(a) Estimate independently all the neural effects ($\tau_{Z_1}$,$\tau_{Z_2}$,$\tau_{Z_3}$); select the strongest ($\tau_{Z_1}$).

(b) Re-estimate the remaining neural effects controlling for the previous discovery ($Z_1$); select the strongest ($\tau_{Z_3}$).

(c) Iterate until no significant neural effects are measured, and return all the selected neurons ( S = $\{Z_1, Z_3\}$).

Figure 4: **Illustrative example of the Neural Effect Search** (NES) recursive procedure with two latent variables $Y_1$, $Y_2$ affected by the treatment and three measurement channels $Z_1$, $Z_2$, $Z_3$, e.g., SAE codes, with $Z_1$ principally aligned with $Y_1$ and $Z_3$ with $Y_2$ (see principal alignment formal definition in Assumption A.2). At each iteration stratifying over the already selected neurons, e.g., $Z_1$ after the first iteration, acts as a proxy controlling for the mediated effects by the corresponding (true) latent effects already identified, i.e., $Y_1$.

---

**Theorem 4.1** (Consistency of Neural Effect Search). *Suppose the outcome neural representation, i.e., SAE codes, captures and almost disentangles the $r$ treatment effects (still allowing for broad effect leakage). Then, as $n \to \infty$, the NES' output $S_{final}$ satisfies*

$$\Pr\Big( S_{final} = \{j_1, \ldots, j_r\}\Big) \longrightarrow 1,$$

*where each $j_k$ is a neuron coordinate principally aligned with a distinct affected outcome $Y_k$.*

---

*Proof Sketch.* At the first round, entanglement makes several coordinates look affected; nevertheless, the coordinate most aligned with some true direction $v_k$ maximizes the treatment effect and, under Bonferroni control, is selected with probability $\to 1$ as $n \to \infty$. Next, (pooled) stratification removes the contribution of the discovered direction from every coordinate: (i) its leakage into other neurons averages out in expectation, and (ii) the post-treatment conditioning transmitting collider bias get bounded. Consequently, all remaining adjusted test statistics are mean-zero (up to vanishing error). By repeating this argument, each iteration peels off one undiscovered principal direction until all $r$ are recovered; thereafter no coordinate exhibits a nonzero mean effect and the procedure halts. Hence $\Pr\big( S_{final} = \{j_1, \ldots, j_r\}\big) \to 1$ and $\mathbb{E}[|S_{final}|] \to r$. Further efficiency results, without loosing consistency, can be obtained with arm-wise effect residualization by the already selected neurons. See Appendix A for the complete formulation, proof and hypotheses discussion. $\square$

**Discussion.** NES recovers the $r$ effect concepts in probability and terminates, in sharp contrast with the paradox described earlier. While standard multi-hypothesis tests collapse with increasing power, i.e., $n$ and $\tau$, proposing *all* entangled neurons with $Y$ as significant effects, NES avoids this pitfall by recursively stratifying. Each iteration removes the spurious signal caused by leakage from already identified effects, and bounding the collider bias, only the unidentified effect factor remains detectable. In this sense, NES does not merely test for effects: it *disentangles* the effects (under Assumptions A.1-A.3), identifying and adjusting the estimation of one true treatment effect factor at a time until the entire effect subspace is recovered. Thus, NES can be interpreted both as a multiple-testing correction method robust to entanglement and as a principled effect disentanglement algorithm.

## 5 RELATED WORKS

**Interpretable Heterogeneous Treatment Effect Estimation.** A closely related line of work is the *empirical* discovery of treatment effect heterogeneity across covariates $W$. Methods such as causal trees, forests, and decision rules ensembles (Athey and Imbens, 2016; Athey et al., 2019; Bargagli-Stoffi et al., 2020) identify subpopulations with different responses, recognizing that pointwise estimation of the Conditional Average Treatment Effect (CATE) is almost impossible to test, and still difficult and risky to interpret. Since $W$ is lower-dimensional, interpretability of these partitions or rules is crucial, and the field has developed around making this empirical exploration scientifically meaningful. Our work is analogous in spirit: instead of asking *who* is affected (heterogeneity over $W$), we ask *what* is affected (discovering affected outcomes $Y$) when the outcome space itself is high-dimensional and initially unknown.

**Causal abstractions and representations.** In the line of work of causal abstractions, Visual Causal Feature Learning (VCFL, Chalupka et al., 2014) was introduced to discover interventions in data rather than outcomes. In scientific trials, however, treatments are fixed by design, and the challenge is to recover their effects from complex outcome measurements. Causal Feature Learning (CFL, Chalupka et al., 2017) extends this to outcome clustering by grouping micro- to macro-variables using equivalence classes of $P(X \mid do(T))$. This requires density estimation in high-dimensional spaces, which is generally infeasible. While clustering other metrics may be possible, CFL commits to a single grouping rule, while we find all statistically significant ones. Another line of work tackles the discovery of causal variables from high-dimensional observations (Schölkopf et al., 2021). Closest in spirit to our setting are interventional approaches (Varici et al., 2023; 2024; Zhang et al., 2023; Yao et al., 2025), which, even with all the necessary extra assumptions, would only offer identification results for the experimental settings $W$ and not the outcome (i.e., the component invariant to the intervention (Yao et al., 2025)). Therefore, they can not be applied to exploratory causal inference because they cannot discover outcome variables.

**Scientific discovery via SAEs.** A related line of work uses SAEs to decompose *polysemantic* hidden representations in foundation models into more *monosemantic* units that align with single concepts (Bricken et al., 2023; Templeton et al., 2024; Huben et al., 2024; Papadimitriou et al., 2025). Although SAEs were initially proposed as an interpretability tool (Bricken et al., 2023), a growing body of negative results, including spurious interpretability on random networks (Heap et al., 2025), failures to isolated atomic concepts (Leask et al., 2025; Chanin et al., 2025), and limited downstream benefits (Wu et al., 2025), casts doubt on whether SAE features faithfully reflect underlying mechanisms rather than post-hoc artifacts. Despite these interpretability concerns, recent work shows that SAEs can still be useful for generating scientific hypotheses from high-dimensional data (Peng et al., 2025). For example, *HypotheSAEs* (Movva et al., 2025) leverage SAEs to surface human-understandable patterns correlated with target outcomes (e.g., engagement levels), which researchers can then treat as hypotheses for follow-up study. Our setting is related but distinct: whereas these approaches focus on correlations and do not provide statistical procedures to test the significance of the unsupervised discoveries, we target *causal* effects and develop inference to assess which high-dimensional outcomes $Y$ are affected, offering principled support for exploratory causal claims.

## 6 EXPERIMENTS

We validate our analyses (significance collapse paradox, and NES consistency) in two complementary settings: a semi-synthetic benchmark where ground-truth causal effects are known, and a real-world randomized trial from experimental ecology.

### 6.1 SEMI-SYNTHETIC BENCHMARK

We simulated a family of RCTs $\{T_i, W_i, Y_i\}_{i=1}^n$, relating both the individual covariates and outcomes one-to-one with specific attributes in the CelebA (Liu et al., 2018) dataset, e.g., `wearing_hat` and `eyeglasses`, and then assigned a random image $X_i$ from the dataset perfectly matching such attributes. Given the corresponding random sample $\{T_i, X_i\}_{i=1}^n$ we (i) trained a SAE over the image representations encoded by SigLIP (Zhai et al., 2023), and (ii) tested NES for effect discovery against vanilla statistical tests ($t$-test, FDR (Schweder and Spjøtvoll, 1982), Bonferroni) and top-$k$ effects selection. For quantitative evaluation, we first assessed SAE monose-

manticity with respect to the considered CelebA attributes (see Figure 8), and extracted the ground truth neurons referring to $Y$. Then, for each effect discovery, we computed Recall, Precision, and Intersection over Union (IoU) with respect to them. Full details about the data generating procedure, training, and evaluation are reported in Appendix D, and the main results ($r = 2$) are summarized in Figure 5. Additional experiments testing (i) the required assumptions, (ii) method consistency with extensive ablations, and (iii) additional baselines are reported in Appendix E.

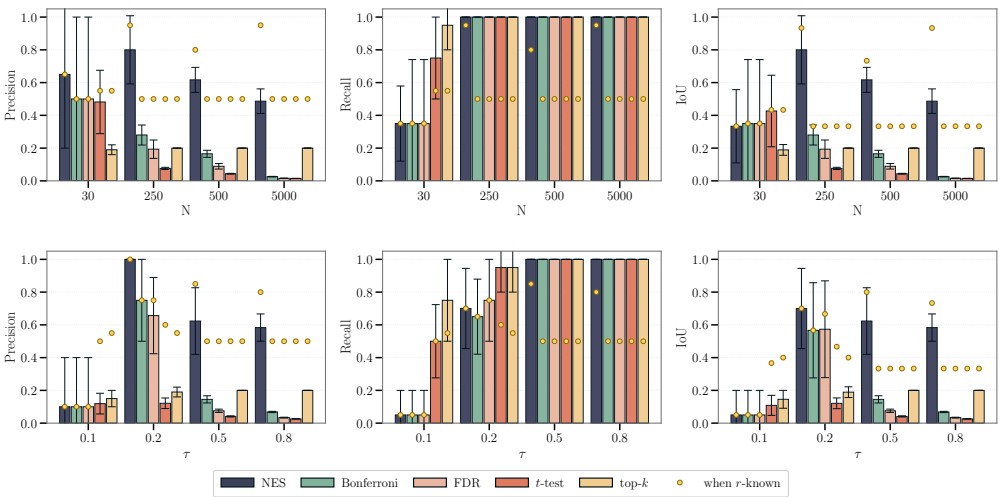

Figure 5: **Semi-synthetic benchmark.** Precision, Recall, and IoU of different testing procedures across sample size $N$ (top) and effect size $\tau$ (bottom). NES consistently achieves the best trade-off, avoiding the significance collapse of standard corrections.

**Results.** Increasing the power of the tests (increasing the sample size $n$ or effect magnitude $\tau$), all the methods eventually retrieve the true significant effects, i.e., Recall $\rightarrow 1$. However, while all the baselines drop the Precision and corresponding IoU (Paradox of Exploratory Causal Inference), NES is the only method that mitigates such entanglement biases. As expected, the Paradox doesn't emerge with very small sample ($n = 30$) and effect regime ($\tau = 0.1$), and more explorative approaches, as vanilla $t$-test or top-$k$ selection, could be preferred, at the price of potentially more false significant hypotheses, i.e., Precision $\ll 1$. With a yellow dot, we report the performance of each method assuming the number of affected outcomes $r$ is known. NES still manages to find both effects most of the time. Instead, all the baseline methods fail to reach Precision and Recall above $0.5$: they succeed in retrieving the most significant effect (equivalent to the first step in NES), but then get confounded by the entanglement and miss the second one. While this is clearly a toy experiment, this is undesirable. For example, if in real trials there are multiple effects with different magnitudes (e.g., the positive effect of a drug on the health metric of interest and rare side effects) the leakage of strong effects may prevail over the weaker ones, which would then be missed.

## 6.2 REAL-WORLD RANDOMIZED TRIAL FROM EXPERIMENTAL ECOLOGY

ISTANT (Cadei et al., 2024) is an ecological experiment where ants from the same colony are randomly exposed to a treatment or a control substance and continuously filmed in triplets in a closed environment to study the concept of Social Immunity. The biologists are interested in identifying which latent behaviors are significantly affected by treatment. According to previous analysis, we first encoded each frame in the trial with DINOv2 (Oquab et al., 2023), and then we trained a SAE directly on the trial data. NES is then applied without Bonferroni adjustment due to the small sample size ($n = 44$ videos) to discover treatment-sensitive codes, and only two neurons are returned.

**Results.** Figure 6 shows a qualitative interpretation of such neurons, by filtering the most and least activated clips in the dataset. In agreement with the previous analysis on the dataset (Cadei et al., 2024; 2025), the first neuron retrieved (`code 394`) seems to separate *grooming* events, already measured as significantly affected by the treatment in any previous rationalist approach to the experiment, i.e., actually manually annotating and testing for it. Quantitatively, such a neuron is exactly the most predictive neuron for grooming event (F1-*score*= 0.398) out of all the 4 608 SAE's codes,

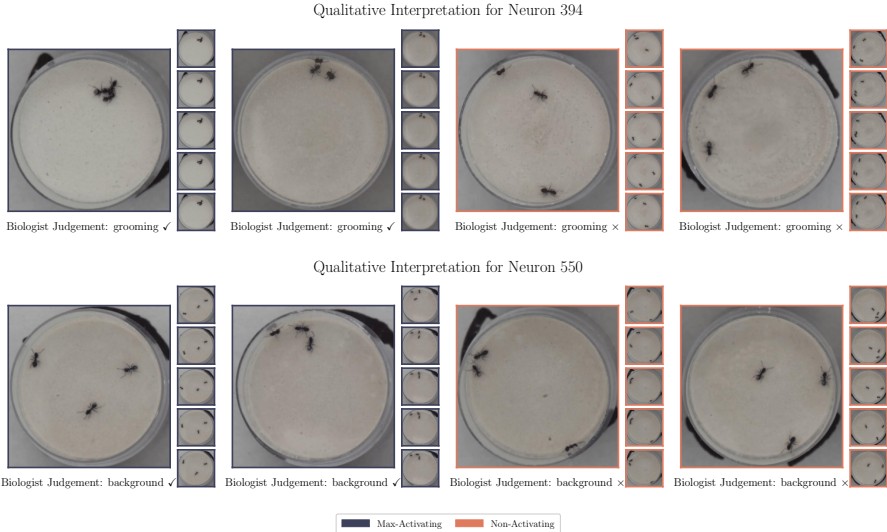

Figure 6: **Exploratory Causal Inference for Experimental Ecology.** Without any knowledge of the behaviors of interest, our procedure retrieves two significant treatment effects, interpreted as (i) grooming behavior and (ii) finite sample experimental design error, in full agreement with previous literature.

confirming the consistent results of our pipeline. The imperfect F1-$score < 1$ suggests the possibility of other entangled effects or broader representation, prohibiting linking the treatment effect on the interpreted outcome with the effect on such a neuron without further labeling. The second neuron activated (`code 550`) represents a black position marking in the background, i.e., a specific black color marking for dish palette positioning on the background (top right outline). Indeed, all the videos recorded with such visible position marking (top left dish palette in the first 4 batches of recordings) are always assigned to the same treatment by chance due to the small experiment size, as discussed in the annotation bias by Cadei et al. (2024). Quantitatively, it is exactly the most predictive neuron for such a characteristic (F1-$score = 0.568$). The fact that the model also identifies the effect of the treatment assignment on the background due to the small sample size is a strength of the method: it is still a statistically significant signal and should not be ignored, even if not interesting directly as a biological effect. It is indeed the domain experts' duty to select which signal is scientifically relevant or just useful to eventually (not necessarily) improve the experiment design.

## 7    CONCLUSION

In this paper, we have discussed how foundation models and SAEs can address the challenges of exploratory causal inference, serving as learned measurement devices. A key challenge is that SAE neurons may not map one-to-one onto high-level concepts, and even weak or mixed associations propagate the dependency on the treatment. This means that many neurons can be activated, making the interpretation difficult as they do not encode a single concept, and they activate more with larger sample sizes or stronger effects. We address this issue with Neural Effect Search, a statistical hypothesis testing procedure that iteratively controls for the biased dependency between neurons after they have been selected. Our experiments on semi-synthetic and real-world randomized trials are encouraging: our method uncovers both scientifically relevant effects and, when present, interpretable finite sample associations, e.g., background marking, that experts can readily dismiss. Overall, we view this work as a first step toward AI-driven efficiency gains in exploratory data science, where foundation models can "look at massive amounts of data first" and then domain experts can identify which patterns have scientific value.

**Our approach has several limitations**. First, we strongly assume that the observed variables $X$ adequately capture information about the unknown $Y$, i.e., data sufficiency. For example, shrinkage in a tumor is detectable via X-ray imaging, but, depending on the tumor type, a treatment may also reduce its metabolic activity, which is measured with PET scans. More complex measurement processes for $X$, e.g., multi-modal are a natural extension of our work. Furthermore, we assume

foundation models encode concepts linearly and that SAEs can approximately recover the effects. We believe the linear representation hypothesis (Park et al., 2023) is mild: even if current foundation models are imperfect, future iterations are likely to improve. The good "*identifiability*" assumption is our strongest. Promising early work already exists (Cui et al., 2025; Hindupur et al., 2025), but the identifiability theory of SAEs is not currently as well understood as that of causal representations (Yao et al., 2025), e.g., still unclear how to deal with continuous concepts. In our paper, we took a more empirical and future-looking stance on improvements in SAEs, focusing on inevitable finite samples entanglement while leveraging pretrained foundation models. Lacking identifiability means that domain experts can today only use our method "as a rescue system for hypotheses they may have missed", before properly annotating the data and following the rationalist approach. We hope that our work will serve as a practical motivation for future work on identifiability in foundation model representations and SAE.

## ACKNOWLEDGMENTS

We thank the Causal Learning and Artificial Intelligence group at ISTA for the continuous feedback on the project and valuable discussions. We further acknowledge the Fourth Bellairs Workshop on Causal Representation Learning held at the Bellairs Research Institute, February 14/21, 2025, and particularly the discussions with Chandler Squires, Dhanya Sridhar, Jason Hartford, and Frederick Eberhardt and further feedback from Judea Pearl. Riccardo Cadei is supported by a Google Research Scholar Award and a Google Google-initiated gift to Francesco Locatello. Tommaso Mencattini was supported by the ISTernship program. This research was funded in part by the Austrian Science Fund (FWF) 10.55776/COE12). It was further partially supported by the ISTA Interdisciplinary Project Committee for the collaborative project "ALED" between Francesco Locatello and Sylvia Cremer. For open access purposes, the author has applied a CC BY public copyright license to any author-accepted manuscript version arising from this submission.

## ETHICS STATEMENT

All datasets used in this work are publicly available. In particular, the ISTAnt dataset (Cadei et al., 2024) was annotated and pre-processed by domain experts. Despite our method guarantees identification under untestable representation learning assumption (without supervision), any conclusion should not be interpreted as scientifically relevant without a domain expert's interpretation. Additionally, since we cannot guarantee identifiability, it should only be used as a rescue system for hypotheses that may have been missed before committing to the rationalist approach, which is anyway necessary for inference, i.e., treatment effect estimation and uncertainty quantification.

## REPRODUCIBILITY STATEMENT

All the datasets we use are publicly available and experiment details are thoroughly detailed in Appendix D-E. A snipped Python implementation for the main algorithm is also presented in Appendix C.

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

# Appendix

# A PROOFS

## A.1 SIGNIFICANCE LEVEL COLLAPSE WITH SAMPLE SIZE (THEOREM 3.1)

**Theorem A.1** (Significance level collapse with sample size). *Let $Z \in \mathbb{R}^m$ be SAE codes and $\tau_j$ the treatment effect on neuron $j$. By definition*

$$\mathcal{A}_\varepsilon := \{\, j : |\tau_j| \geq \varepsilon \,\}, \qquad |\mathcal{A}_\varepsilon| = \rho_\varepsilon m. \tag{7}$$

*In multiple testing at level $\alpha$, regardless of Bonferroni correction*

$$\Pr\Big(all \ j \in \mathcal{A}_\varepsilon \ are \ rejected\Big) \ \to \ 1 \quad as \ n \to \infty, \tag{8}$$

*and the number of rejections $R_n$ satisfies*

$$R_n \ \to \ \rho_\varepsilon m \quad in \ probability. \tag{9}$$

*In words: as the sample size grows, all entangled neurons with the (true) affected outcomes are declared significantly affected by the treatment, regardless of being principally related to other concepts.*

*Proof.* For each neuron $j$, let $\hat\tau_j$ be the estimated treatment effect and $t_j$ its $t$-statistic. Under standard randomization, we have the asymptotic distribution

$$t_j \ \xrightarrow{d} \ \mathcal{N}(\lambda_j, 1), \qquad \lambda_j = \tfrac{\sqrt{n}}{\sigma} \tau_j, \tag{10}$$

where $\sigma^2$ is the asymptotic variance of $\hat\tau_j$.

Multiple testing with Bonferroni adjustment rejects $H_{0j} : \tau_j = 0$ if $|t_j| > z_{\alpha/(2m)}$, where $z_{\alpha/(2m)}$ is the $(1 - \alpha/(2m))$ quantile of $\mathcal{N}(0,1)$. As $m \to \infty$, the threshold satisfies

$$z_{\alpha/(2m)} \ \asymp \ \sqrt{2 \log m}. \tag{11}$$

For any $j \in \mathcal{A}_\varepsilon$, we have $\tau_j \neq 0$, hence $\lambda_j$ diverges at rate $\sqrt{n}$ as $n \to \infty$. Since $\sqrt{n}$ grows faster than $\sqrt{\log m}$, it follows that

$$\Pr\big(|t_j| > z_{\alpha/(2m)}\big) \ \to \ 1. \tag{12}$$

Therefore, for all $j \in \mathcal{A}_\varepsilon$, the null is rejected with probability tending to one, and analogously

$$\Pr\big(|t_j| > z_{\alpha/2}\big) \ \to \ 1. \tag{13}$$

without Bonferroni adjustment. By the union bound,

$$\Pr\Big(\text{all } j \in \mathcal{A}_\varepsilon \text{ are rejected}\Big) \ \to \ 1. \tag{14}$$

Hence, the number of rejections converges in probability to $|\mathcal{A}_\varepsilon| = \rho_\varepsilon m$, proving the claim. $\square$

## A.2 SIGNIFICANCE COLLAPSE WITH EFFECT MAGNITUDE (COROLLARY 3.2)

**Corollary A.1** (Significance collapse with effect magnitude). *Fix a finite sample size $n$. Suppose the treatment effects scale as*

$$\tau_j(s) = s\,\gamma_j, \qquad j = 1, \ldots, m, \tag{15}$$

*where $s > 0$ is a scaling parameter and $\gamma_j$ are fixed coefficients. By definition*

$$\mathcal{A}_\varepsilon := \{\, j : |\gamma_j| > \tfrac{\varepsilon}{s} \,\}, \qquad |\mathcal{A}_\varepsilon| = \rho_\varepsilon m. \tag{16}$$

*In multiple testing at level $\alpha$ regardless of the Bonferroni correction,*

$$\Pr\Big(all \ j \in \mathcal{A}_\varepsilon \ are \ rejected\Big) \ \to \ 1 \quad as \ s \to \infty, \tag{17}$$

*and the number of rejections $R_s$ satisfies*

$$R_s \ \to \ \rho_\varepsilon m. \quad in \ probability. \tag{18}$$

> *In words: even at a fixed sample size, amplifying the effect magnitude all the entangled neurons with the (true) affected outcomes are declared significantly affected by the treatment, regardless of being principally related to other concepts.*

*Proof.* For neuron $j$, the noncentrality parameter of the $t$-statistic under effect scaling $s$ is

$$\lambda_j(s) = \tfrac{\sqrt{n}}{\sigma}\,\tau_j(s) = \tfrac{\sqrt{n}}{\sigma}\,s\gamma_j. \tag{19}$$

If $\gamma_j = 0$, then $\lambda_j(s) = 0$ for all $s$ and the rejection probability remains bounded by $\alpha/m$.

If $\gamma_j \neq 0$, then $\lambda_j(s) \to \infty$ linearly in $s$, while the Bonferroni threshold $z_{\alpha/(2m)}$ is fixed (since $n, m$ are fixed). Therefore,

$$\Pr\big(|t_j| > z_{\alpha/(2m)}\big) \;\to\; 1 \quad \text{as } s \to \infty. \tag{20}$$

Analogously, without Bonferroni $\tfrac{1}{m}$ significance correction. Thus, for every $j \in \mathcal{A}_\varepsilon$, the null is eventually rejected with probability tending to one. By independence of limits,

$$\Pr\Big(\text{all } j \in \mathcal{A}_\varepsilon \text{ are rejected}\Big) \;\to\; 1, \tag{21}$$

and $R_s \to \rho_\varepsilon m$ in probability, completing the proof. $\square$

## A.3 CONSISTENCY OF NEURAL EFFECT SEARCH (THEOREM 4.1)

Given a Randomized Controlled Trial, with randomized treatment $T \in \{0, 1\}$, (unobserved) affected outcome $Y \in \mathbb{R}^r$, i.e., with non-null effect, and the SAE codes $Z \in \mathbb{R}^m$ characterizing each individual/observation extracted from the indirect outcome measurements $X \in \mathcal{X} \subseteq \mathbb{R}^{p}$[1] By design (RCT), we furthermore assume SUTVA and finite second moments with standard Lindeberg regularity within strata. Let the average treatment effect on the (ground truth) outcome be:

$$\tau^Y := \mathbb{E}[Y \mid do(T{=}1)] - \mathbb{E}[Y \mid do(T{=}0)] \in \mathbb{R}^r, \tag{22}$$

and the average treatment effect on the SAE codes:

$$\tau^Z := \mathbb{E}[Z \mid do(T{=}1)] - \mathbb{E}[Z \mid do(T{=}0)] \in \mathbb{R}^m, \tag{23}$$

then,

$$\tau^Z \;=\; \sum_{k=1}^{r} \tau_k^Y\, v_k \;=\; V\,\tau^Y, \tag{24}$$

where the matrix $V = [v_1 \;\cdots\; v_r] \in \mathbb{R}^{m \times r}$ aggregates in columns the $r$ affected outcome neural representations (see Definition in Section 3).

**Assumption A.1** (Sufficiency). *The matrix of code-level effect directions $V$ has full column rank, i.e.,* $\mathrm{rank}(V) = r$.

*Discussion.* By the neural treatment effect decomposition, i.e., Equation 24, every code-level contrast lies in $\mathrm{span}\{v_1, \dots, v_r\}$. Assumption A.1 guarantees this span is truly $r$-dimensional (nondegenerate) and that the linear map $\tau^Y \mapsto \tau^Z$ is injective—distinct effect vectors $\tau^Y$ produce distinct code-level contrasts. Informally, at the *mean-effect* level, this behaves like "no loss of information" about $\tau^Y$ when passing through $V$ (akin to a sufficient statistic for $\tau^Y$ in a parametric family).

**Assumption A.2** (Alignment). *There exist* distinct *indices* $j_1, \dots, j_r \in [m]$ *such that*

$$|v_{k,j_k}| \;>\; \max_{\ell \neq k} |v_{\ell,j_k}| \qquad \forall k \in [r], \tag{25}$$

*each effect direction $v_k$ has a distinct principal neuron $j_k$ that strictly dominates the other effect directions by* max. *In addition, the following* global Principal–Max *property holds for the realized effect vector $\tau^Y$:*

$$\max_{j \in [m]} \Big| \sum_{\ell=1}^{r} \tau_\ell^Y\, v_{\ell j} \Big| \;=\; \max_{k \in [r]} \Big| \sum_{\ell=1}^{r} \tau_\ell^Y\, v_{\ell,\, j_k} \Big|. \tag{26}$$

---

[1]As a special case, the following arguments also hold for binary affected outcomes and neuronal representations in SAE.

*Discussion.* Equation 25 supplies a distinct principal neuron (geometric dominance by max) per affected outcome factor. The global Principal–Max condition 26 states that, in population, the overall argmax of the code-level contrast $\tau^Z = V\tau^Y$ is attained at *some* principal neuron. Because each NES round removes discovered effects from the sum, replacing $\sum_{\ell=1}^{r}$ in Equation 26 by a subsum over the remaining (undiscovered) indices only reduces nonprincipal candidates; hence the argmax remains principal at every round.

**Assumption A.3** (Arm-wise Effect Decomposition with $\varepsilon$-Leakage). *For any NES round $\ell$ with discovered set $S_{\ell-1} := S$ and any $j \notin S$,*

$$\mathbb{E}[Z_j \mid Z_S, \, do(T{=}t)] \;=\; h_{j,t}(Z_S) \;+\; \rho_{j,t}, \qquad t \in \{0,1\}, \tag{27}$$

*where $h_{j,t}$ is measurable w.r.t. $\sigma(Z_S)$ ($\sigma$-algebra), and*

$$\rho_{j,1} - \rho_{j,0} \;=\; \sum_{k \notin K(S)} \tau_k^Y \, v_{k,j}, \tag{28}$$

*with $K(S)$ the affected outcome components already identified by $S$. Moreover, let $G := g(Z_S)$ be the pooled (treatment-agnostic) stratification. Define the arm/stratum discrepancy*

$$\Delta_{j,t}(g) := \mathbb{E}[Z_j \mid G{=}g, \, T{=}t] - \mathbb{E}[Z_j \mid G{=}g, \, do(T{=}t)].$$

*with vanishing transport error, after all affected outcome components are identified (i.e., when $K(S) = \{1, \dots, r\}$):*

$$\Delta_{j,t}(g) = 0 \quad \text{for all } j \notin S, \; g \in \mathcal{G}, \; t \in \{0,1\} \tag{29}$$

*There exists a constant $\varepsilon \geq 0$ such that, for the weights $w_g$ used by the estimator,*

$$\left| \sum_g w_g \Big( \Delta_{j,1}(g) - \Delta_{j,0}(g) \Big) \right| \;\leq\; \varepsilon \qquad \text{for all } j \notin S. \tag{30}$$

*Finally, to ensure that principal coordinates remain identifiable in the presence of leakage, assume the (population) principal margin*

$$\Gamma \;:=\; \max_{k \in [r]} \left| \tau_k^Y \, v_{k,j_k} \right| \;-\; \max_{j \notin \{j_1, \dots, j_r\}} \left| \sum_{\ell=1}^{r} \tau_\ell^Y \, v_{\ell j} \right| \tag{31}$$

*satisfies $\Gamma > 2\varepsilon$.*

*Discussion.* Assumption A.3 is a *mean-level* conditional requirement: conditioning within arms on the already-selected codes $Z_S$ explains their contribution in expectation, leaving only undiscovered effects in the adjusted contrast. In addition, Equation 30 is a mild *pooled-strata transport* bound: even though $G$ is post-treatment and may induce a collider opening, the *weighted* difference between observed and interventional arm means within strata is uniformly bounded by $\varepsilon$ and vanishes. In the linear (or locally linear) regime, the Jacobian of the mapping $Y \mapsto Z$ satisfies $\partial Z / \partial Y = V$ (or more generally $\partial\, \mathbb{E}[Z \mid Y, do(T)] / \partial Y = V$), so the columns $v_k$ are Jacobian columns; Assumption A.2 requires principal dominance, while Assumption A.3 ensures additive mean structure and bounded transport error so that stratification cancels already-discovered parts up to a uniform $\varepsilon$ bias.

> **Theorem.** *Under randomization, SUTVA, finite second moments with Lindeberg regularity, and Assumptions A.1–A.3, NES with pooled stratification on $Z_S$ and Bonferroni control selects one new direction per round and halts after $r$ rounds with probability $\to 1$, identifying all the principal components, i.e.,*
>
> $$\Pr\big(S_{\text{final}} = \{j_1, \dots, j_r\}\big) \to 1, \qquad \mathbb{E}[|S_{\text{final}}|] \to r.$$

*Note.* Since the Theorem establishes $\Pr\big(S_{\text{final}} = \{j_1, \dots, j_r\}\big) \to 1$, it follows directly that also the overall family-wise error rate (FWER) and false discovery rate (FDR) of the entire recursive NES procedure converge to 0.

We proceed by proving NES consistency, decomposing it in the following four steps:

1. average neural treatment effect identification by randomization and SUTVA, i.e., RCT,
2. neural effect estimation by *pooled* stratification is unbiased up to a uniform $\varepsilon$ and, by Assumption A.3, cancels the contribution of already-discovered directions up to $\varepsilon$, leaving only the undiscovered part,
3. at each round the largest adjusted coordinate identifies, i.e., principal neuron, a new affected outcome by Assumption A.2 and is detected under standard CLT scaling,
4. by Assumption A.1, after $r$ rounds no adjusted mean contrast remains beyond $\varepsilon$ and NES halts.

**Step 1:** Average neural treatment effect identification by randomization.

**Proposition A.1** (Average Neural Treatment Effect Identification). *For each coordinate $j$,*

$$\tau_j^Z = \mathbb{E}[Z_j \mid do(T{=}1)] - \mathbb{E}[Z_j \mid do(T{=}0)] = \mathbb{E}[Z_j \mid T{=}1] - \mathbb{E}[Z_j \mid T{=}0]. \quad (32)$$

*Proof.* Randomization implies $P(Z \mid do(T{=}t)) = P(Z \mid T{=}t)$. Taking expectations and using SUTVA yields Equation 32. □

**Step 2:** Neural Effect estimation by pooled stratification: bounded bias and disentanglement.

At round $\ell = 1$, $\mathtt{S} = \varnothing$, there is no stratification, and the corresponding associational difference identifies the average treatment effect by standard RCT results (shown in Proposition A.1). At round $\ell > 1$, build strata $\mathcal{G}$ by deterministically binning $Z_{\mathtt{S}}$ (pooled quantile cutpoints, ignoring $T$). For each stratum $g \in \mathcal{G}$ and arm $t \in \{0, 1\}$, let $\overline{Z}_{j,tg}$ denote the sample mean of $Z_j$ among units with $(T = t, G = g)$, and $n_{tg}$ their count.

For each $j \notin \mathtt{S}$, define the post-stratified estimator

$$\widehat{\tau}_j^{\text{strat}} = \sum_{g \in \mathcal{G}} w_g \big( \overline{Z}_{j,1g} - \overline{Z}_{j,0g} \big), \qquad w_g \propto n_{1g} + n_{0g}, \quad (33)$$

see Algorithm 2 for weight definition. Across $j \notin \mathtt{S}$, use Bonferroni level $\alpha/m$ and add the top-$|\widehat{\tau}_j^{\text{strat}}|$ rejection.

**Lemma A.1** (Neural Effect estimation by pooled stratification: bounded bias and disentanglement). *Let $\widehat{\tau}_j^{\text{strat}}$ be defined by Equation 33. Then*

$$\mathbb{E}\big[\widehat{\tau}_j^{\text{strat}}\big] = \sum_{g \in \mathcal{G}} \Pr(G{=}g)\big(\mathbb{E}[Z_j \mid G{=}g, do(T{=}1)] - \mathbb{E}[Z_j \mid G{=}g, do(T{=}0)]\big) + B_j, \quad (34)$$

*where the leakage bias*

$$B_j := \sum_{g \in \mathcal{G}} w_g \Big( \Delta_{j,1}(g) - \Delta_{j,0}(g) \Big)$$

*satisfies $|B_j| \leq \varepsilon$ by Equation 30. Under Assumption A.3, the contribution explained by $Z_{\mathtt{S}}$ averages out within arm in the interventional means, giving*

$$\left| \mathbb{E}\big[\widehat{\tau}_j^{\text{strat}}\big] - \sum_{k \notin K(\mathtt{S})} \tau_k^Y v_{k,j} \right| \leq \varepsilon. \quad (35)$$

*Moreover, under finite second moments and Lindeberg regularity, the $t$-statistic of $\widehat{\tau}_j^{\text{strat}}$ is asymptotically normal with variance consistently estimated by the usual post-stratified Neyman formula.*

*Proof.* Add and subtract $\mathbb{E}[Z_j \mid G{=}g, do(T{=}t)]$ inside each arm/stratum mean to obtain Equation 34 and identify $B_j$. Assumption A.3 yields Equation 27 and Equation 28, so averaging over $G$ cancels the $h_{j,t}(Z_{\mathtt{S}})$ part and preserves the stratum-invariant $\rho_{j,t}$, whose contrast equals Equation 28; combining with $|B_j| \leq \varepsilon$ gives Equation 35. The CLT follows from standard post-stratified difference-in-means theory with finite second moments and nonvanishing stratum proportions. □

**Step 3:** NES one-step correctness.

**Proposition A.2** (One-step correctness). *Suppose the discovered set $\mathsf{S}$ retrieves exactly the principal neurons for the $K(\mathsf{S})$ affected outcome factors already identified. Then for any $j \notin \mathsf{S}$,*

$$\left| \mathbb{E}[\widehat{\tau}_j^{\text{strat}}] - \sum_{k \notin K(\mathsf{S})} \tau_k^Y \, v_{k,j} \right| \le \varepsilon.$$

*By Assumption A.2 and the margin condition $\Gamma > 2\varepsilon$ in Assumption A.3, the largest adjusted coordinate identifies, i.e., principal neuron, a new affected outcome and is rejected, i.e., selected as significantly affected by the treatment, with probability $\to 1$ as $n \to \infty$.*

*Proof.* By Lemma A.1, the adjusted mean at $j$ equals the sum over undiscovered directions up to $\pm\varepsilon$. Fix an undiscovered $k^\star$ and its principal coordinate $j_{k^\star}$ given by Assumption A.2. The principal margin $\Gamma > 2\varepsilon$ ensures that the principal signal at $j_{k^\star}$ dominates the maximal nonprincipal signal by more than $2\varepsilon$, hence remains strictly largest after a $\pm\varepsilon$ perturbation. Its $t$-statistic diverges at rate $\sqrt{n}$, so the maximizer over $j \notin \mathsf{S}$ is a true undiscovered coordinate with probability $\to 1$. $\qquad\square$

**Step 4:** NES consistency by induction.

**Theorem** (NES Consistency). *Under randomization, SUTVA, finite second moments with Lindeberg regularity, and Assumptions A.1–A.3, NES with pooled stratification on $Z_S$ and Bonferroni control selects one new direction per round and halts after $r$ rounds with probability $\to 1$, identifying all the principal neurons, i.e.,*

$$\Pr\big(\mathsf{S}_{\text{final}} = \{j_1, \ldots, j_r\}\big) \to 1, \qquad \mathbb{E}[|\mathsf{S}_{\text{final}}|] \to r.$$

*Proof.* We distinguish between no affected outcomes, i.e., $r = 0$ and at least one, i.e., $r \ge 1$.

- If $r = 0$ (trivial), then $\tau^Y = 0$ and by Equation 24 also $\tau^Z = 0$. Hence all adjusted expectations are 0, no coordinate is selected at any round, and $\mathsf{S}_{\text{final}} = \varnothing$ with probability $\to 1$ as $n \to \infty$.

- If $r \ge 1$ with $\tau_k^Y \ne 0$ for each $k \in \{1, \ldots, r\}$:

  *Base step.* This is the special case of Proposition A.2 with $\mathsf{S} = \varnothing$: the largest neural effect identifies a principal direction and is rejected with probability $\to 1$ as $n \to \infty$.

  *Induction step.* Suppose at the beginning of round $\ell$ ($1 < \ell \le r$) the discovered set $\mathsf{S}$ identifies exactly the $\ell - 1$ distinct principal neurons identifying $K(\mathsf{S})$. By Lemma A.1,

  $$\left| \mathbb{E}\big[\widehat{\tau}_j^{\text{strat}}\big] - \sum_{k \notin K(\mathsf{S})} \tau_k^Y \, v_{k,j} \right| \le \varepsilon \qquad \text{for every } j \notin \mathsf{S},$$

  i.e., the adjusted mean at any candidate coordinate depends only on undiscovered directions up to a uniform $\varepsilon$ bias. Pick any undiscovered $k^\star \notin K(\mathsf{S})$ and its principal coordinate $j_{k^\star}$ (Assumption A.2 still applies to the remaining columns). By the margin condition $\Gamma > 2\varepsilon$, the associated $t$-statistic diverges and the maximizer over $j \notin \mathsf{S}$ is tied to an undiscovered direction with probability $\to 1$.

  *Termination.* Each round adds one previously undiscovered direction with probability $\to 1$. By Assumption A.1 ($\text{rank}(V) = r$), there are exactly $r$ linearly independent effect directions; after $r$ rounds they are all represented, and Lemma A.1 together with the final-stage exactness equation 29 yields

  $$\mathbb{E}\big[\widehat{\tau}_j^{\text{strat}}\big] = 0 \qquad \text{for every remaining } j.$$

  Hence, the corresponding $t$-statistics are $O_p(1)$ and, under Bonferroni control, no hypothesis is rejected with probability $\to 1$. Thus, no further selections occur, and $\Pr\big(\mathsf{S}_{\text{final}} = \{j_1, \ldots, j_r\}\big) \to 1$ and $\mathbb{E}[|\mathsf{S}_{\text{final}}|] \to r$.

*Comment.* If a pre-treatment effect modifier $W$ influences the codes used to stratify (i.e., $W \to Z_{\mathcal{S}}$), pooled conditioning can create transport discrepancies; in that case, if some additional SAE code outside $\mathcal{S}$ captures (part of) this modifier, the same margin argument ensures it will be selected, enlarge $\mathcal{S}$, and—by Equation 29—the discrepancy collapses thereafter. Conversely, if no such modifier projects into the stratification variables (i.e., $W \not\to Z_{\mathcal{S}}$), then $\Delta_{j,t}(g) \equiv 0$ already and no leakage arises.

$\square$

**Convergence Rate**  The proof of Theorem 4.1 shows that, at each round and for each remaining principal neuron $j_k$, the corresponding $t$-statistic has a noncentrality parameter of order $\sqrt{n}$. In particular, the smallest principal margin that NES can detect with nontrivial power scales as $O(n^{-1/2})$, as in standard CLT-based tests.

**Residualization (*optional*).**  The theory above uses pooled stratification on $Z_{\mathrm{S}}$ with the $\varepsilon$-leakage bound Equation 30. In practice, one may *arm-wise residualize* the tested coordinate $Z_j$ on $Z_{\mathrm{S}}$ (e.g., by OLS within each arm) and then apply the same stratified contrast, or use a plain arm difference. This targets the same estimand and can improve finite-sample power, but it is not strictly required.

*Why it helps (variance reduction).* Fix an arm $t$ and consider the $L^2(P(\cdot \mid T{=}t))$ projection $R_{j,t} := Z_j - \beta_t^\top Z_{\mathrm{S}}$ with $\beta_t = \arg\min_\beta \mathbb{E}[(Z_j - \beta^\top Z_{\mathrm{S}})^2 \mid T{=}t]$. Then

$$\mathrm{Var}(R_{j,t} \mid T{=}t) = \min_\beta \mathrm{Var}(Z_j - \beta^\top Z_{\mathrm{S}} \mid T{=}t) \;\leq\; \mathrm{Var}(Z_j \mid T{=}t).$$

Thus, for any stratification weights, the usual Neyman variance for the difference in means built on $R_{j,t}$ is weakly smaller asymptotically than that built on $Z_j$ (componentwise, within strata). Intuitively, residualization orthogonalizes $Z_j$ against already-discovered codes within the arm, removing predictable variation and shrinking the standard error.

*Why it can hurt (finite-sample and misspecification effects).* If $\beta_t$ is estimated (say $\hat{\beta}_t$) on the same samples used to test $j$, two issues arise: (i) **Estimation noise** can inflate variance when $\mathrm{S}$ is large or collinear, partially offsetting the variance reduction above. (ii) **Signal leakage** in finite samples: although the *population* projection preserves the undiscovered mean contrast under Assumption A.3, an overfitted $\hat{\beta}_t$ can inadvertently subtract some of the undiscovered mean component at $j$ (attenuating the signal and reducing power). Both effects vanish asymptotically if $\hat{\beta}_t \to \beta_t$.

*Validity and a safe recipe.* If residualization is performed *within arm* and $\beta_t$ is estimated on *independent folds* (sample-splitting/cross-fitting), then

$$\mathbb{E}[\overline{R}_{j,tg}] \;=\; \mathbb{E}[R_{j,t} \mid G{=}g, do(T{=}t)] + o(1),$$

and the Step 2 bounded-bias/cancellation proof applies with $R_j$ in place of $Z_j$. Hence, residualization is asymptotically valid and (typically) more efficient. In small samples without splitting, we still target the same estimand in expectation up to $o_p(1)$ under standard regularity, but power can be non-monotone due to estimation noise.

*Recommendation.* We suggest using arm-wise residualization as a complementary efficiency device, with cross-fitting (or estimating $\beta_t$ on previously discovered rounds) to avoid overfitting. It cannot worsen asymptotic validity, often improves power by reducing variance, and—implemented with splitting—helps reduce the practical impact of the bounded leakage $\varepsilon$ in Equation 30.

# B  NEURAL EFFECT TEST

---

**Algorithm 2** Neural Effect Test (NET) with stratification on arm-wise residuals

---

1: **function** NEURALEFFECTTEST($T$, $Z$, $j$, S)
2:    // **A) Arm-wise residualize only the tested neuron** $j$ (*optional*)
3:    **if** S $= \varnothing$ **then**
4:        set $r_j \leftarrow Z_{\cdot j}$   (*first round: no residualization*)
5:    **else**
6:        **for** $t \in \{0, 1\}$ **do**
7:            regress $Z_{\cdot j}$ on $Z_{\cdot, S}$ using only samples with $T = t$
8:            for each $i$ with $T_i = t$: $r_{j,i} \leftarrow Z_{ij} - \hat{\beta}_t^{(j)\top} Z_{i, S}$
9:        **end for**
10:    **end if**

11:    // **B) Stratification from raw** $Z_{\mathbf{S}}$
12:    **if** S $= \varnothing$ **then**
13:        put all units in a single stratum: $\mathcal{G} = \{\text{all}\}$   (*first round: no stratification*)
14:    **else**
15:        compute pooled (ignore $T$) medians/quantiles of each $Z_{\cdot s}$, $s \in$ S
16:        assign each unit $i$ to a cell $g(i)$ by binning $Z_{i, S}$ via those cutpoints
17:        drop any stratum $g$ with $n_{1g} = 0$ or $n_{0g} = 0$
18:    **end if**

19:    **for** each stratum $g \in \mathcal{G}$ **do**
20:        $n_{1g}, n_{0g} \leftarrow$ treated/control counts in $g$
21:        $\mu_{1g}, \mu_{0g} \leftarrow$ treated/control means of $r_j$ in $g$
22:        $\sigma_{1g}^2, \sigma_{0g}^2 \leftarrow$ treated/control variances of $r_j$ in $g$
23:        $w_g \leftarrow \dfrac{n_{1g} + n_{0g}}{\sum_h (n_{1h} + n_{0h})}$
24:    **end for**
25:    $\hat{\tau}_j \leftarrow \sum_g w_g (\mu_{1g} - \mu_{0g})$
26:    $V \leftarrow \sum_g w_g^2 \left( \dfrac{\sigma_{1g}^2}{n_{1g}} + \dfrac{\sigma_{0g}^2}{n_{0g}} \right)$
27:    $t \leftarrow \dfrac{\hat{\tau}_j}{\sqrt{V}}$;    $\nu \leftarrow \dfrac{V^2}{\sum_{g \in \mathcal{G}} \left( \dfrac{\left( w_g^2\, \sigma_{1g}^2/n_{1g} \right)^2}{\max(n_{1g} - 1, 1)} + \dfrac{\left( w_g^2\, \sigma_{0g}^2/n_{0g} \right)^2}{\max(n_{0g} - 1, 1)} \right)}$   $\triangleright$ Satterthwaite df
28:    $p \leftarrow 2 \cdot \Pr(|T_\nu| \geq |t|)$   $\triangleright$ tests $H_0 : \tau_j^R = 0$
29:    **return** $(\hat{\tau}_j,\ p)$
30: **end function**

---

The algorithm tests whether neuron $j$ still carries a *residual* causal contrast after accounting for already–discovered effects S. We first compute an *arm-wise* residual $r_j := Z_j - \hat{\beta}_T^{(j)\top} Z_S$, where $\hat{\beta}_t^{(j)}$ is fit using only units with $T = t$. Arm-wise fitting avoids pooled "bad control" on post-treatment codes and cancels leakage from previously found directions as they manifest within each arm.

We then form treatment-agnostic strata $\mathcal{G}$ by coarsening the raw $Z_S$ (e.g., medians/quantiles computed *pooled* over $T$) and drop cells lacking both arms. Within each $g \in \mathcal{G}$ we take the treated–control mean difference of $r_j$ and aggregate with weights $w_g \propto n_{1g} + n_{0g}$. This is standardization (g-computation):

$$\hat{\tau}_j = \sum_g w_g \left( \bar{r}_{j,1g} - \bar{r}_{j,0g} \right) \xrightarrow{\mathbb{E}} \sum_g \Pr(G{=}g)(\mathbb{E}[r_j \mid G, g, do(1)] - \mathbb{E}[r_j \mid G, g, do(0)]) = \tau_j^R,$$

so the estimator is unbiased under randomization/SUTVA. The reported variance and Satterthwaite df are the usual stratified formulas.

## C MINIMAL PYTHON IMPLEMENTATION SNIPPET

We provide a minimal Python snippet for our algorithm relying on `pandas` library (McKinney et al., 2011) for tabular operations and `SciPy` library for statistical testing (Virtanen et al., 2020) .

**Neural Effect Search** Recursive discovery.

```python
def NES(T, Z, S=[], alpha=0.05):
    m = Z.shape[1]
    tests = NET(T, Z, S)
    R = tests.loc[(tests["p_value"] <= alpha/m)]
    if R.empty:
        return S
    j = R["ATE"].abs().idxmax()
    return NES(T, Z, S=S.append(j), alpha=alpha)
```

**Neural effect Test** By stratification with median binning. For simplicity, we ignore here arm-wise residualization (see full code for detailed implementation).

```python
import pandas as pd
import scipy

def NET(T, Z, S=[]):

    # columns to test
    cols = [c for c in Z.columns if c not in set(S)]
    if not cols:
        return pd.DataFrame(columns=["neuron","ATE","p_value"])

    # build strata id
    df = Z.copy()
    df["_T"] = T.astype(int)
    if S:
        # two groups per stratifier (median split); pooled (ignore T)
        # could use also df[s]>0 as alternative
        gid_bits = [(df[s] > df[s].median()).astype(int) for s in S]
        df["_gid"] = pd.concat(gid_bits, axis=1).apply(tuple, axis=1)
    else:
        df["_gid"] = 0

    N = len(df)
    out = []

    # stratify and test each neuron
    for j in cols:
        ATE = 0.0
        Vsum = 0.0
        denom = 0.0

        for g, dg in df.groupby("_gid"):
            n_g = len(dg)
            if n_g < 3:
                continue
            x1 = dg.loc[dg["_T"] == 1, j]
            x0 = dg.loc[dg["_T"] == 0, j]
            n1, n0 = len(x1), len(x0)
            if n1 < 2 or n0 < 2:
                continue
```

```
            # per-stratum summaries
            mu1, mu0 = x1.mean(), x0.mean()
            s1,  s0  = x1.var(ddof=1), x0.var(ddof=1)
            w = n_g / N

            # LOTP aggregation
            ATE += w * (mu1 - mu0)
            V_g = (s1 / n1) + (s0 / n0)
            Vsum += (w**2) * V_g
            denom += (w**4) * ((s1 / n1)**2 / max(n1 - 1, 1) + (s0 / n0)
                **2 / max(n0 - 1, 1))

        se = Vsum**0.5
        tstat = ATE / se
        df_ws = (Vsum**2) / denom
        pval = 2.0 * scipy.stats.t.sf(abs(tstat), df=df_ws)
        out.append((j, float(ATE), float(pval)))

    return pd.DataFrame(out, columns=["neuron","ATE","p_value"])
```

# D EXPERIMENTS DETAILS

## D.1 CELEBA SEMI-SYNTHETIC RCTS

**Dataset.** We use CelebA (Liu et al., 2018), a face attributes dataset with $> 200k$ images and 40 binary attributes per image [2]. Furthermore, for implementation details, labels have been doubled (we pass from `Beard` to `Has_Beard` and `Has_notBeard`). We follow the authors' official *train/-val/test* split, and we employ the validation data for training SAEs and the test data to interpret them. Attributes are treated as ground-truth binary labels. From this source, we simulate several RCTs following the data-generating process (DGP) described below, varying the sample size ($n \ll 200k$) and treatment effect ($\tau$), reflecting realistic randomized controlled trial characteristics.

**Data Generating Processes** To evaluate discovery accuracy with known ground truth, we simulate RCTs by reusing real images but stochastically sampling treatment and outcomes from CelebA attributes:

- *Treatment:* $T \sim \text{Bernoulli}(0.5)$.
- *Outcome factors:* we designate two binary effects $Y=(Y_1, Y_2)$ using CelebA attributes: $Y_1=$`Eyeglasses`, $Y_2=$`Wearing_Hat`.
- *Exogenous Cause:* $W=$`Smiling`.

We implement a "co-effect" model in which an intervention on $T$ shifts both $Y_1$ and $Y_2$ by the same ATE with arm-specific probabilities and $W$ modifies only $Y_1$, i.e.,:

$$\Pr(Y_2{=}1 \mid T{=}1) = p_1^{(Y_2)}, \quad \Pr(Y_2{=}1 \mid T{=}0) = p_0^{(Y_2)},$$

$$\Pr(Y_1{=}1 \mid T{=}t, W{=}w) = \begin{cases} p_{11}^{(Y_1)} & (t{=}1, w{=}1) \\ p_{10}^{(Y_1)} & (t{=}1, w{=}0) \\ p_{01}^{(Y_1)} & (t{=}0, w{=}1) \\ p_{00}^{(Y_1)} & (t{=}0, w{=}0) \end{cases}$$

with $W \sim \text{Bernoulli}(0.5)$. We vary effect magnitude via an ATE grid $\text{ATE} \in \{0, 0.1, \dots, 0.8\}$ (9 values). Concretely, starting from a base rate $0.5$, we set:

$$p_1^{(Y_2)} = 0.5 + \tfrac{\text{ATE}}{2}, \quad p_0^{(Y_2)} = 0.5 - \tfrac{\text{ATE}}{2},$$

and analogously for $Y_1$ in the $W{=}1$ arm:

$$p_{11}^{(Y_1)} = 0.5 + \tfrac{\text{ATE}}{2}, \quad p_{01}^{(Y_1)} = 0.5 - \tfrac{\text{ATE}}{2}, \quad p_{10}^{(Y_1)} = 0.2 + \text{ATE}, \quad p_{00}^{(Y_1)} = 0.2.$$

For each simulated unit, we draw $(T, W, Y_1, Y_2)$, then assign an *actual image* whose CelebA attributes match the realized $(Y_1, Y_2, W)$.

**FM features.** Each image $x$ is encoded with SigLIP (Zhai et al., 2023) into a patch-level representation; we use the final-layer token features (dim $d{=}768$, 196 patches/token positions averaged). Unless noted otherwise, we do not use any task-specific fine-tuning.

**SAE Details.** We train a SAE on SigLIP features to obtain interpretable codes $Z \in \mathbb{R}^m$ that serve as hypotheses for treatment effect estimation. Thereafter, the details for the SAE in Table 1. Lastly, to turn hidden representation into hypotheses, aggregate patchwise by *mean pooling* to a single $Z \in \mathbb{R}^{9216}$ per image. These per-image codes are the units we test in NES and baseline procedures.

**Evaluation.** We evaluate discoveries against concept–aligned SAE codes extracted from CelebA. Let $m{=}9\,216$ be the number of codes and $Z_j(X) \in \mathbb{R}$ the activation of code $j \in [m]$ on image $X$; a code is *active* when $Z_j(X) > 0$. For true effect $Y_k \in \{0, 1\}$ (here $k \in \{1, 2\}$) and each code $j$, we induce predictions $\hat{y}_{ik}^{(j)} := \mathbb{I}\{Z_j(X_i) > 0\}$ and compute the F1-score of $\{\hat{y}_{ik}^{(j)}\}_{i=1}^n$ against the ground-truth labels $\{y_{ik}\}_{i=1}^n$; the *best* neuron for the concept is then

$$g_k := \arg\max_{j \in [m]} \text{F1}\Big(\{\hat{y}_{ik}^{(j)}\}_{i=1}^n, \{y_{ik}\}_{i=1}^n\Big).$$

---

[2] It can be downloaded from flwrlabs/celeba.

| Component | Setting |
|-----------|---------|
| Encoder nonlinearity | Top-$k$ with $k$=5 active codes |
| Input dimension | 768 |
| Code / decoder dimension ($m$) | 9216 |
| Optimizer / LR / batch | Adam / $5 \times 10^{-4}$ / 20 |
| Epochs / grad clipping | 20 / 1.0 |

Table 1: Training details for the SAE employed in semi-synthetic experiments.

The resulting ground-truth set of affected codes is $\mathcal{G} := \{g_1, g_2\}$ (in general $|\mathcal{G}| = r$). Each method (NES or a baseline) returns a set of discovered codes $\mathcal{S} \subseteq [m]$, which we compare to $\mathcal{G}$ via set metrics. Defining $\text{TP} := |\mathcal{S} \cap \mathcal{G}|$, $\text{FP} := |\mathcal{S} \setminus \mathcal{G}|$, and $\text{FN} := |\mathcal{G} \setminus \mathcal{S}|$, we report

$$\text{Precision} = \frac{\text{TP}}{\text{TP} + \text{FP}}, \qquad \text{Recall} = \frac{\text{TP}}{\text{TP} + \text{FN}}, \qquad \text{F1} = \frac{2 \, \text{Precision} \cdot \text{Recall}}{\text{Precision} + \text{Recall}},$$

and the set Intersection-over-Union (IoU)

$$\text{IoU} = \frac{|\mathcal{S} \cap \mathcal{G}|}{|\mathcal{S} \cup \mathcal{G}|} = \frac{\text{TP}}{\text{TP} + \text{FP} + \text{FN}} .$$

For each trained representation, including the additional experiments in Appendix E, we repeat the evaluation resampling the experiment 10 times and reporting the average $\pm$ standard deviation. Further ablations additionally retraining the encoder, i.e., SAE, are presented in Appendix E.

### D.2 ISTAnt

**Data and RCT.** We considered the randomized controlled trial introduced by Cadei et al. (2024). Videos of ant triplets were collected under randomized treatment/control assignment. Throughout our unsupervised pipeline, *domain annotations from biologists were used only a posteriori for interpretation/evaluation of discovered codes, never for training*, as discussed in the main text.

**FM features.** Each frame $X$ is encoded with DINOv2 (Oquab et al., 2023) into a patch-level representation; we use the final-layer token features (dim $d$=384, 256 patches/token positions averaged). Unless noted otherwise, we do not use any task-specific fine-tuning.

**SAE Details.** We train a SAE on the DINOv2 features to obtain interpretable codes $Z \in \mathbb{R}^m$ that serve as hypotheses for treatment effect estimation. Thereafter, the details for the SAE are in Table 2. Lastly, to turn hidden representation into hypotheses, we aggregate patchwise by *mean pooling* to a single $Z \in \mathbb{R}^{4608}$ per frame. These per-frame codes are the units we test in NES and baseline procedures.

| Component | Setting (ISTAnt) |
|-----------|------------------|
| Encoder nonlinearity | top-$k$ with $k$=20 active codes |
| Input dimension | 384 |
| Code / decoder dimension ($m$) | 4608 |
| Optimizer / LR / batch | Adam / $5 \times 10^{-4}$ / 128 |
| Epochs / grad clipping | 10 / 1.0 |

Table 2: Training details for the SAE employed on ISTAnt.

**Evaluation.** Evaluation follows exactly the CELEBA protocol: we score discovered codes against ground-truth concepts via code–induced predictions and compute Precision/Recall/F1 and IoU for the set of returned codes (with domain annotations used only for interpreting and quantifying performance, not for training).

# E    ADDITIONAL EXPERIMENTS

In this Section, we provide extensive empirical validation of our method, in addition to the selected main semi-synthetic experiments reported in Section 6. Particularly,

- we explicitly **test** for the required encoder representation **assumptions**, theoretically justifying the successful identification results;

- we extensively validate the **method consistency** by (i) repeating the experiments with different encoders and trainings, (ii) comparing different method hyper-parameters, and (iii) additionally testing on more varied data-generating processes;

- we additionally include two **novel** heuristic **baselines**, inspired by Causal Feature Learning principles (Chalupka et al., 2017).

Overall, the experiments consistently validate our identification result and further provide practical guidelines on how to set the different hyperparameters.

## E.1    ASSUMPTIONS VALIDATION

We assess how well SAE codes behave as measurement channels on CELEBA by testing the alignment between individual neurons and the ground–truth attributes described in the previous section. We consider the main SAE trained for the main experiments reported in Section 6 with training details described in Appendix D, and we then compare it with the same encoder replacing SigLIP (foundational model) with DINOv2, and again SAE top-$k$ non linearity with JumpRELU. According to Theorem 4.1, we test if (i) such attributes information is still encoded in the SAE representation, i.e., Assumption A.1 and (ii) per each attribute there exists a dedicated neuron strongly aligned and all the other neurons are just marginally aligned with it, i.e., Assumption A.2. We first test the foundational model sufficiency and then for the SAE. Particularly, for each code $j$ in the SAE representation, we treat the event $Z_j > 0$ as a binary predictor and compute its F1–score against each attribute label (assumed given). We then interpret the most predictive neurons by visualizing their corresponding most activated samples; and visualize such attributes' entanglement in the representation by filtering all their most predictive codes and comparing their probing performances. We remark that in practice, without the affected outcome supervision, these assumptions are not testable, and we can do it here just because controlling the data-generating process and having access to attribute annotations. More indirect and unsupervised assessments for a SAE representation, testing a priori validity for NES, should be explored in future works, or just ignored, at the price of having no guarantees on the interpretability of NES results.

**SigLIP with SAE top-$k$ non linearity**    (main encoder). As expected, SigLIP embeddings are sufficient for the task, enabling us to accurately classify both the affected outcomes directly via (vanilla) logistic regression, respectively with Precision$= 0.9639$ and Recall$= 0.9947$ for `Wearing_Hat`, and with Precision$= 0.9712$ and Recall$= 0.9747$ for `Eyeglasses`. Furthermore, in the SAE representation, trained with top-$k$ non-linearity ($k = 20$), the two most predictive neurons for the two affected factors are: (i) neuron `38` for `Wearing_Hat` with $\text{F1} - \text{score} = 0.841$, and (ii) neuron `6051` for `Eyeglasses` with $\text{F1} - \text{score} = 0.748$. We can further appreciate qualitatively the high separations by visualizing the most activated images per neuron in Figure 7. Also, the SAE representation is then sufficient, and for each affected outcome, there is a principal neuron strongly aligned. In Figure 8 we visualize the F1-score spectra of the most predictive neurons for each attribute. The long tail, dominated by a principal component, represents exactly what is already required by Assumptions A.1-A.2 to enforce NES validity. Such low–amplitude but widespread correlations are precisely what triggers instead the *Paradox of Exploratory Causal Inference* with vanilla multiple-testing, i.e., with enough power all such entangled neurons are returned as significantly affected due to their (marginal) true affected outcome information. Our NES counters this issue by retrieving the leading effect first and then recursively stratifying on previously discovered codes, so that subsequent tests target the *residual* causal signal rather than previous causes' information leakages.

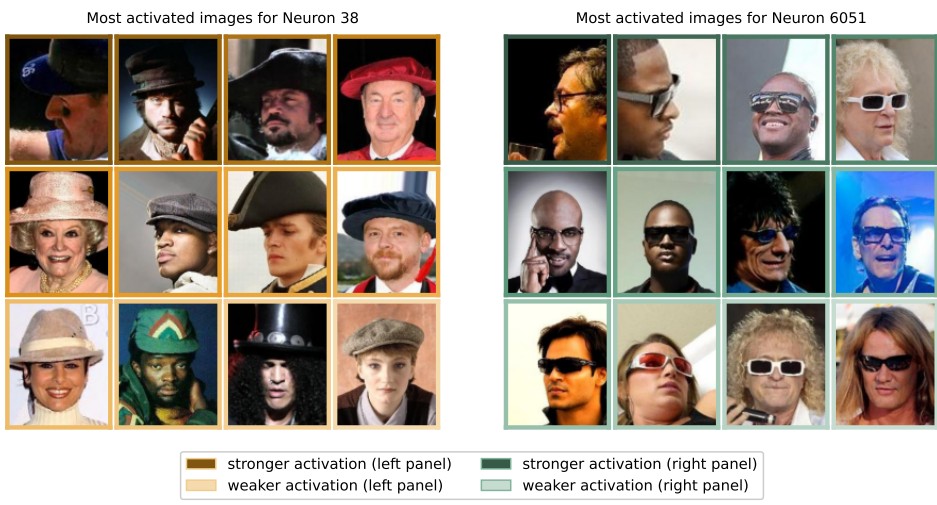

Figure 7: **Qualitative neurons' interpretations** (SigLIP + SAE top-$k$). Each panel shows the 12 most–activated test images for the most predictive neuron of each affected outcome concept (activation = highest code value). Both concepts are perfectly separated.

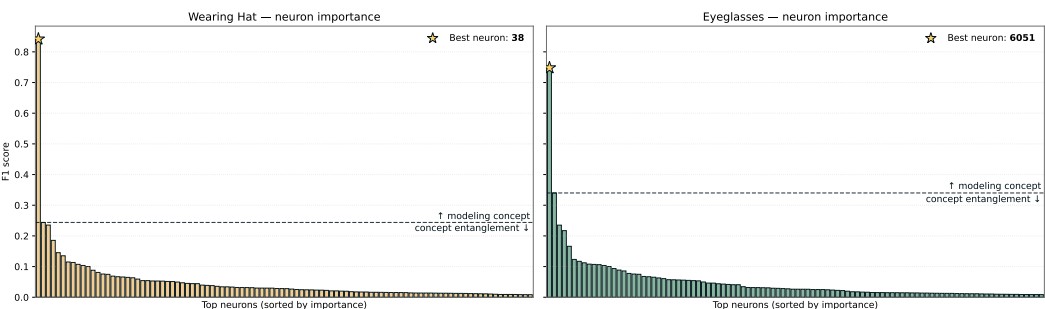

Figure 8: **Assessing principal alignment with broad entanglement** (SigLIP + SAE top-$k$). For each attribute, we rank SAE codes by F1-score against the affected attributes and visualize the most predictive in order. Both concepts are principally explained by a single neuron, and still weakly, while broadly entangled with many others.

**DINOv2 with SAE top-$k$ non-linearity** As expected, also DINOv2 embeddings are sufficient for the task, enabling to accurately classify both the affected outcomes directly via (vanilla) logistic regression, respectively with Precision= 0.9534 and Recall= 0.9787 for `Wearing_Hat`, and with Precision= 0.9636 and Recall= 0.9567 for `Eyeglasses`. Furthermore, in the SAE representation, trained with top-$k$ non-linearity ($k = 20$), the two most predictive neurons for the two affected factors are: (i) neuron `1937` for `Wearing_Hat` with F1 − score = 0.431, and (ii) neuron `253` for `Eyeglasses` with F1 − score = 0.594. We can further appreciate qualitatively the separations by visualizing the most activated images per neuron in Figure 9. Also, the SAE representation is then sufficient, while the principal alignment is high for `Eyeglasses` but less evident for `Wearing_Hat`, where the concepts could be possibly captured by all the top three most predictive neurons. Indeed, also in the qualitative interpretation, among the most activated samples, some concepts, e.g., wearing sunglasses and a hat, are over-represented. In Figure 10, we visualize the F1-score spectra of the most predictive neurons for each attribute. The long tail, dominated by a principal component, represents exactly what is already required by Assumptions A.1-A.2 to enforce NES validity, even if the principal alignment is weakly satisfied by `Wearing_Hat` concept (potentially violating Equation 26). Such low–amplitude but widespread correlations are precisely what trigger the *Paradox of Exploratory Causal Inference* with vanilla multiple-testing, i.e., with enough power, all such entangled neurons are returned as significantly affected due to their (marginal) true

affected outcome information. Our NES counters this issue by retrieving the leading effect first and then recursively stratifying on previously discovered codes, so that subsequent tests target the *residual* causal signal rather than previous causes' information leakages.

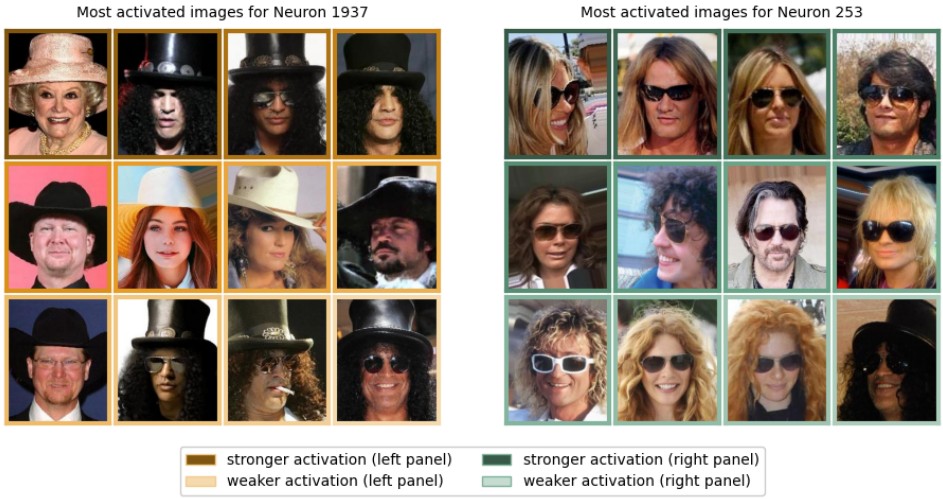

Figure 9: **Qualitative neurons' interpretations** (DINOv2 + SAE top-$k$). Each panel shows the 12 most–activated test images for the most predictive neuron of each affected outcome concept (activation = highest code value). Both concepts are perfectly separated, despite `Wearing_Hat` seeming to over-represent specific subgroups.

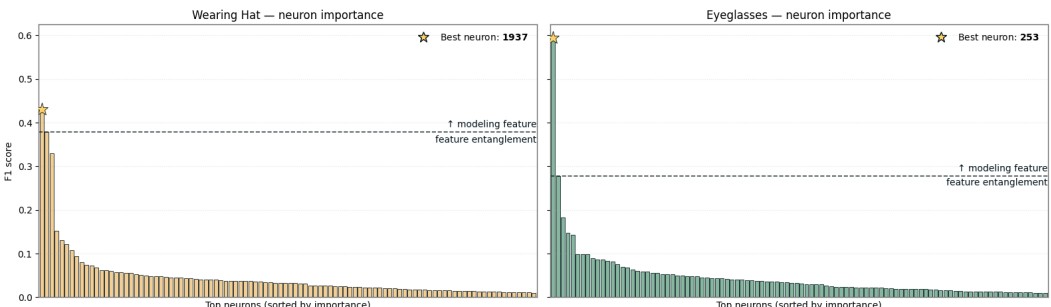

Figure 10: **Assessing principal alignment with broad entanglement** (DINOv2 + SAE top-$k$). For each attribute, we rank SAE codes by F1-score against the affected attributes and visualize the most predictive in order. `Eyeglasses` is principally explained by a single neuron, `Wearing_Hat` by three main neurons, and both concepts are still weakly but broadly entangled with many others.

**SigLIP with SAE JumpReLU non-linearity** We already discussed SigLIP embedding sufficiency in the main encoder hypotheses assessment. However, training on top a SAE with jump-ReLU non-linearity breaks the desiderata hypotheses satisfied with top-$k$ non-linearity. Particularly, the two most predictive neurons for the two affected factors are: (i) neuron `3485` for `Wearing_Hat` with $F1 - score = 0.273$, and (ii) neuron `2865` for `Eyeglasses` with $F1 - score = 0.425$. Both performances are limited, particularly for `Wearing_Hat` classification. We can further appreciate qualitatively the weak separability by visualizing the most activated images per neuron in Figure 11, e.g., Neuron `3485` is not capturing precisely the concept `Wearing_Hat` but probably something broader like "*having something on top of the head*", e.g., some text from the background. In Figure 10, we visualize the F1-score spectra of the most predictive neurons for each attribute. For `Eyeglasses`, there exists a principal neuron, despite the other entangled neurons still being sufficiently predictive. Instead, for `Wearing_Hat` classification, there is no clear principal component, violating NES assumption, and also ill-defining any evaluation attempts (requiring setting the princi-

pal neurons identifying the effects). On such a representation, NES has no guarantees, and the long and heavy tails also trigger once again the *Paradox of Exploratory Causal Inference* with vanilla multiple-testing.

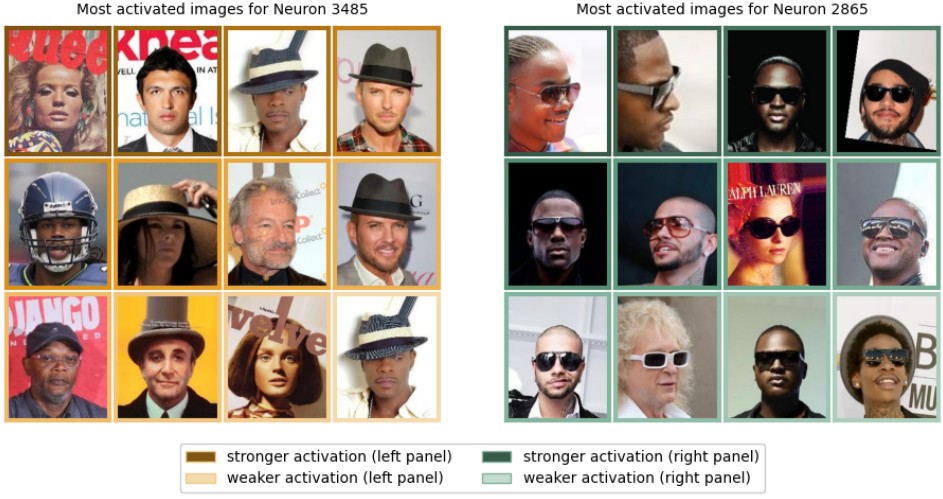

Figure 11: **Qualitative neurons' interpretations** (SigLIP + SAE jump-ReLU). Each panel shows the 12 most–activated test images for the most predictive neuron of each affected outcome concept (activation = highest code value). `Eyeglasses` is separated while it is not the case for `Wearing_Hat` concept.

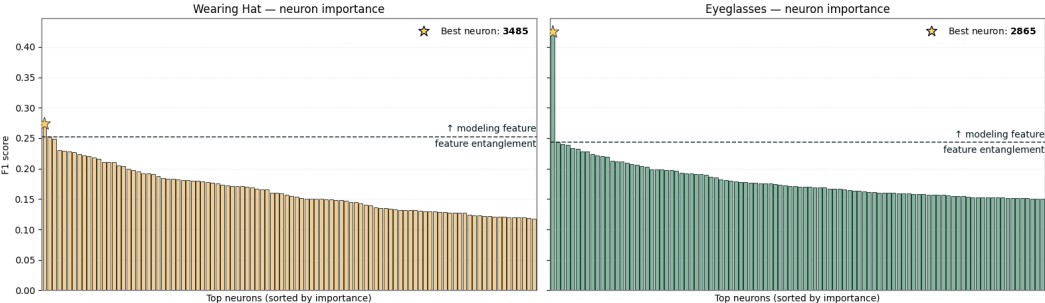

Figure 12: **Assessing principal alignment with broad entanglement** (SigLIP + SAE jump-ReLU). For each attribute, we rank SAE codes by F1-score against the affected attributes and visualize the most predictive in order. `Eyeglasses` concept is principally explained by a single neuron, and still strongly and broadly entangled with many others. `Wearing_Hat` concept has no principal components and the tails are very heavy, i.e., the information is shared among different neurons.

## E.2 METHOD CONSISTENCY

In this Section, we extensively validate the method consistency beyond the experiments selected for the main results discussion in Section 6. Particularly, (i) we repeat the experiments with different encoders and trainings, (ii) we compare different method hyper-parameters, and (iii) we additionally test on more varied data-generating processes.

### E.2.1 VARYING ENCODER

Considering the same data-generating processes and evaluations described in Appendix D, we repeat the experiments relying on different measurement channels, i.e., input encoders and corresponding embedding.

**Foundational model** First, we replicate the main experiment replacing SigLIP with DINOv2 (Oquab et al., 2023) as foundational model. From the experiment in the previous paragraph, we know that the main hypotheses for NES to work hold, still triggering the Paradox of Causal Inference if relying otherwise on vanilla multiple hypothesis testing. The results are reported in Figure 13. As expected, DINOv2 perfectly replicates the results discussed in Section 6.

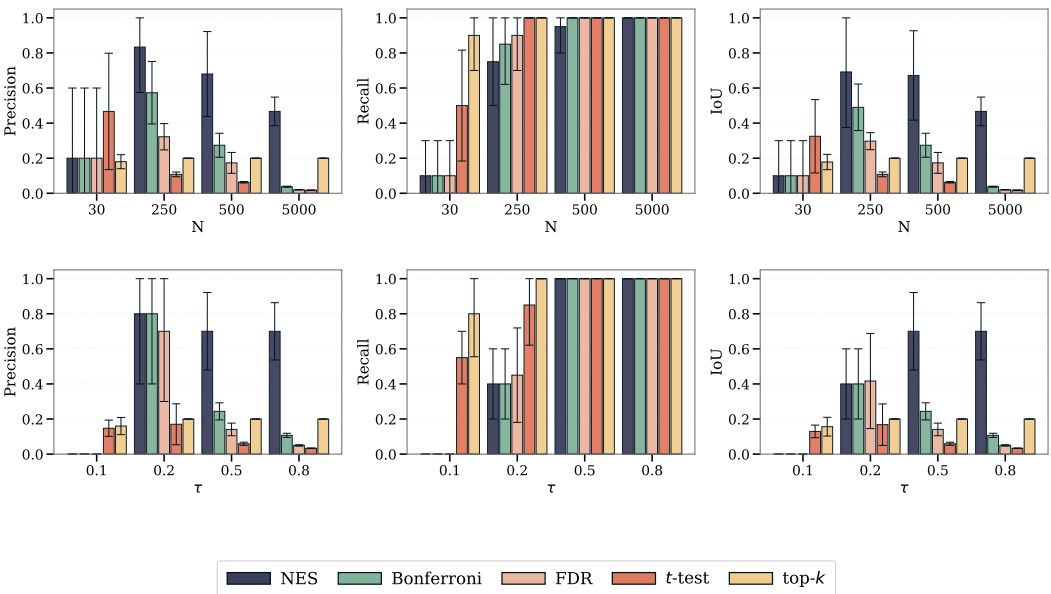

Figure 13: **Ablation Foundationa Model** (DINOv2). Precision, Recall, and IoU in effect identification varying sample size $N$ (top) and effect size $\tau$ (bottom), relying on DINOv2 embeddings. NES consistently replicates the main results using SigLIP embeddings, still overcoming the paradox of Exploratory Causal Inference.

**Sparse autoencoder dimension** Then, we consider different sparse autoencoders varying the latent dimension $m$, and retrying with the same setting described in Appendix D. Particularly, we varied $m$ by a factor $\{4, 6, 8, 10, 14, 16\}$ of the foundational model dimension $d = 768$ (in the main experiment, we considered $m = 12 \cdot d$). In all the settings and retraining, we replicate the results discussed in Section 6, i.e., NES consistently overcomes the Paradox of Exploratory Causal Inference, while it consistently invalidates all the other baselines, making both our method and the baselines invariant with respect to the latent representation dimension, in the order of a few thousand neurons. See results summaries in Figures 14-19.

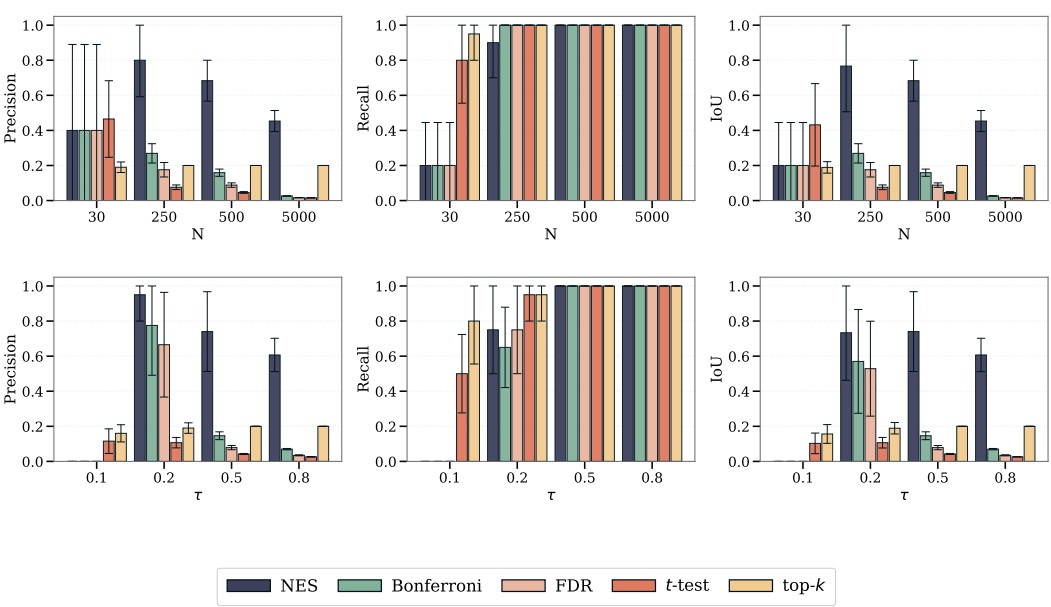

Figure 14: **Ablation SAE dimension** ($m = 3\,072$). Precision, Recall, and IoU in effect identification varying sample size $N$ (top) and effect size $\tau$ (bottom), with SAE dimension $m = 3\,072$, 4x DINOv2 dimension $n = 768$. NES consistently replicates the main results, varying the representation dimension $m$, still overcoming the paradox of Exploratory Causal Inference.

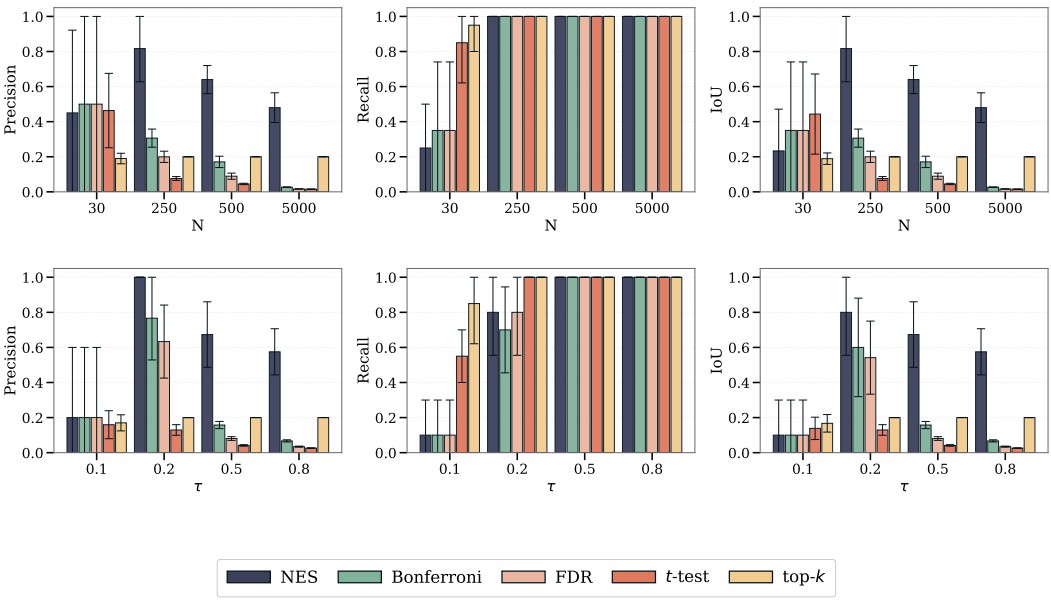

Figure 15: **Ablation SAE dimension** ($m = 4\,608$). Precision, Recall, and IoU in effect identification varying sample size $N$ (top) and effect size $\tau$ (bottom), with SAE dimension $m = 4\,608$, 6x DINOv2 dimension $n = 768$. NES consistently replicates the main results, varying the representation dimension $m$, still overcoming the paradox of Exploratory Causal Inference.

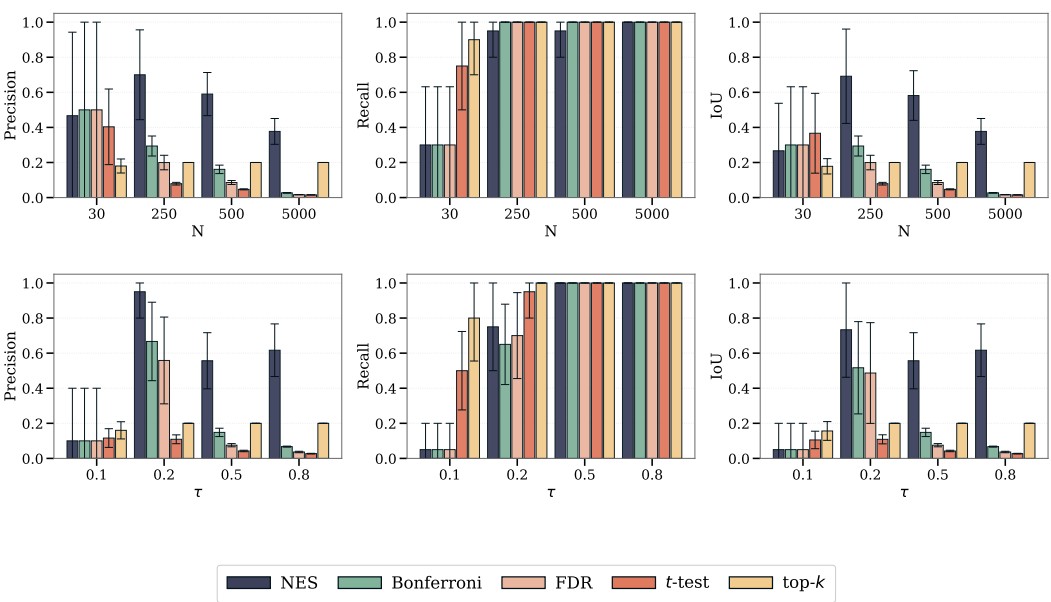

Figure 16: **Ablation SAE dimension** ($m = 6\,144$). Precision, Recall, and IoU in effect identification varying sample size $N$ (top) and effect size $\tau$ (bottom), with SAE dimension $m = 6\,144$, 8x DINOv2 dimension $n = 768$. NES consistently replicates the main results, varying the representation dimension $m$, still overcoming the paradox of Exploratory Causal Inference.

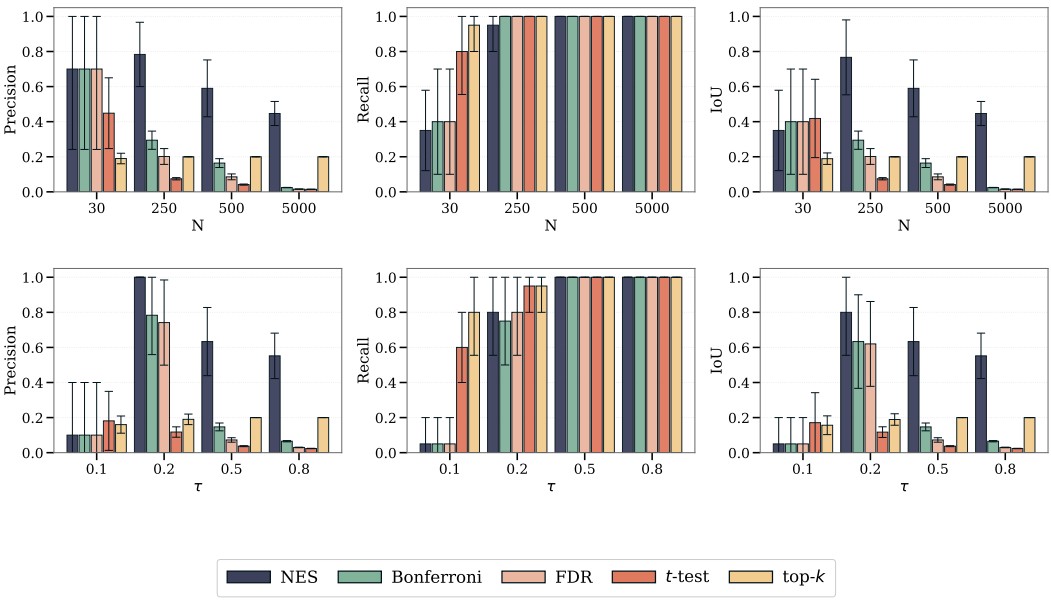

Figure 17: **Ablation SAE dimension** ($m = 7\,680$). Precision, Recall, and IoU in effect identification varying sample size $N$ (top) and effect size $\tau$ (bottom), with SAE dimension $m = 7\,680$, 10x DINOv2 dimension $n = 768$. NES consistently replicates the main results, varying the representation dimension $m$, still overcoming the paradox of Exploratory Causal Inference.

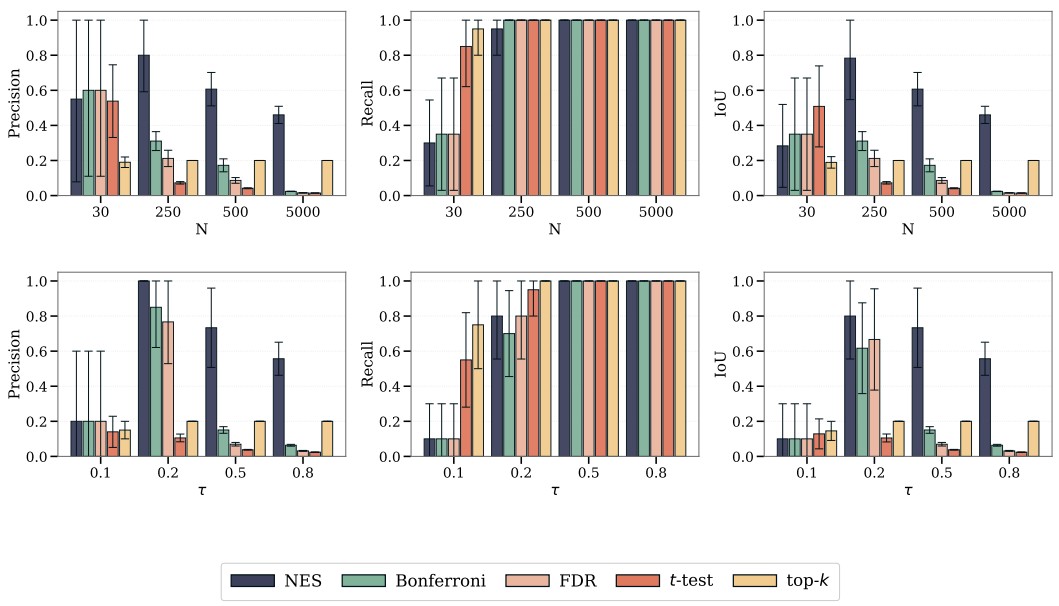

Figure 18: **Ablation SAE dimension** ($m = 10\,752$). Precision, Recall, and IoU in effect identification varying sample size $N$ (top) and effect size $\tau$ (bottom), with SAE dimension $m = 10\,752$, 14x DINOv2 dimension $n = 768$. NES consistently replicates the main results, varying the representation dimension $m$, still overcoming the paradox of Exploratory Causal Inference.

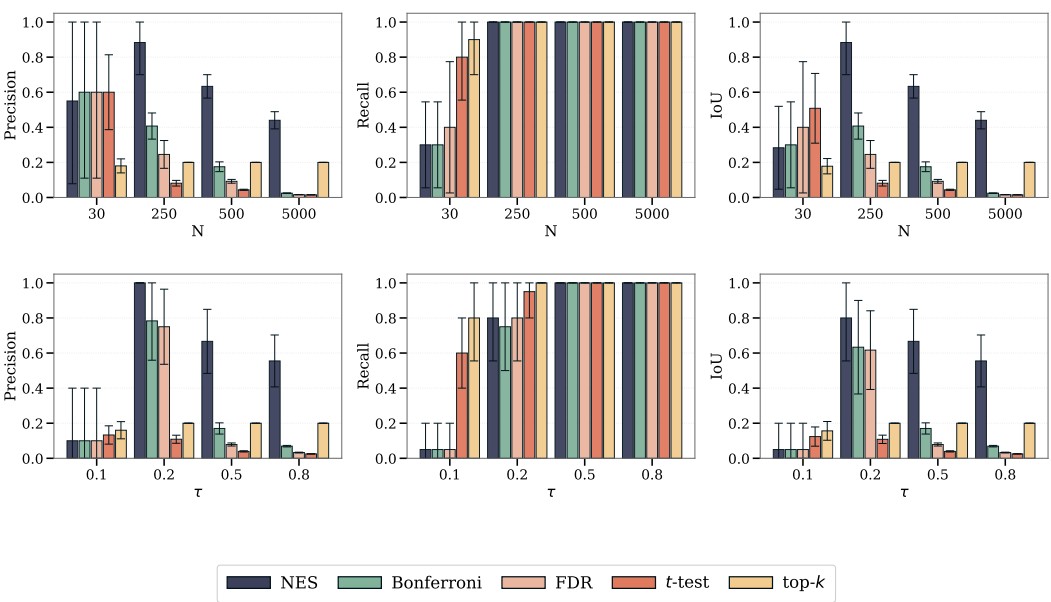

Figure 19: **Ablation SAE dimension** ($m = 12\,288$). Precision, Recall, and IoU in effect identification varying sample size $N$ (top) and effect size $\tau$ (bottom), with SAE dimension $m = 12\,288$, 16x DINOv2 dimension $n = 768$. NES consistently replicates the main results, varying the representation dimension $m$, still overcoming the paradox of Exploratory Causal Inference.

**Sparse autoencoder non-linearity**    Additionally, we consider different sparse auto-encoder non-linearities and repeat all the experiments as before. In Figures 20-23 we report all the results using top-$k$ activation with $k \in \{5, 10, 50, 100\}$ (by default $k = 20$ in the main experiments). Again,

all the results discussed in Section 6 are replicated, i.e., NES consistently overcomes the Paradox of Exploratory Causal Inference, suggesting the top-$k$ selection is not influential for our method, i.e., the required assumptions are consistently satisfied, despite the obtained representation may significantly change. On the other hand, the other baselines consistently fail due to the Paradox of Exploratory Causal Inference, and the higher the enforced entanglement, i.e., $k$, the higher the broad entanglement with the affected outcomes, and accordingly smaller the precision. Finally, any analysis using jump-ReLU for non-linearity is invalid due to the principal alignment discussed above, required both for NES validity and more generally for valid evaluation. For completeness, we report anyway in Figure 24 the replicated experiments results using jump-ReLU non-linearity. At least one, if not both, ground truths are ill-defined, and the consistent null precision, even for NES, should be taken with a pinch of salt.

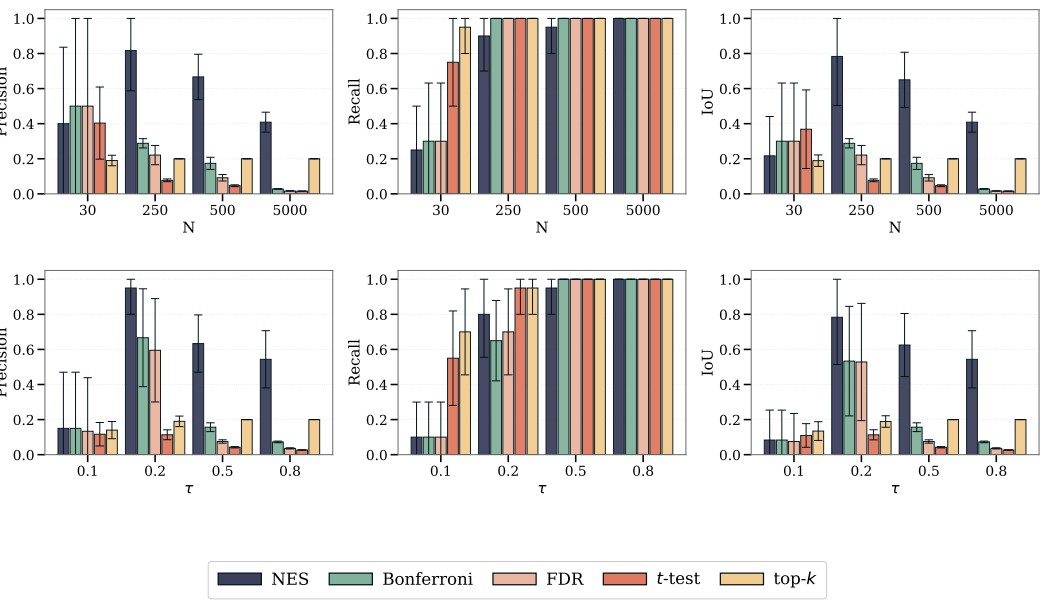

Figure 20: **Ablation SAE non-linearity** (top-5). Precision, Recall, and IoU in effect identification varying sample size $N$ (top) and effect size $\tau$ (bottom), with SAE non-linearity by top-$k$ with $k = 5$. NES consistently replicates the main results using top-$k$ with $k = 20$, still overcoming the paradox of Exploratory Causal Inference.

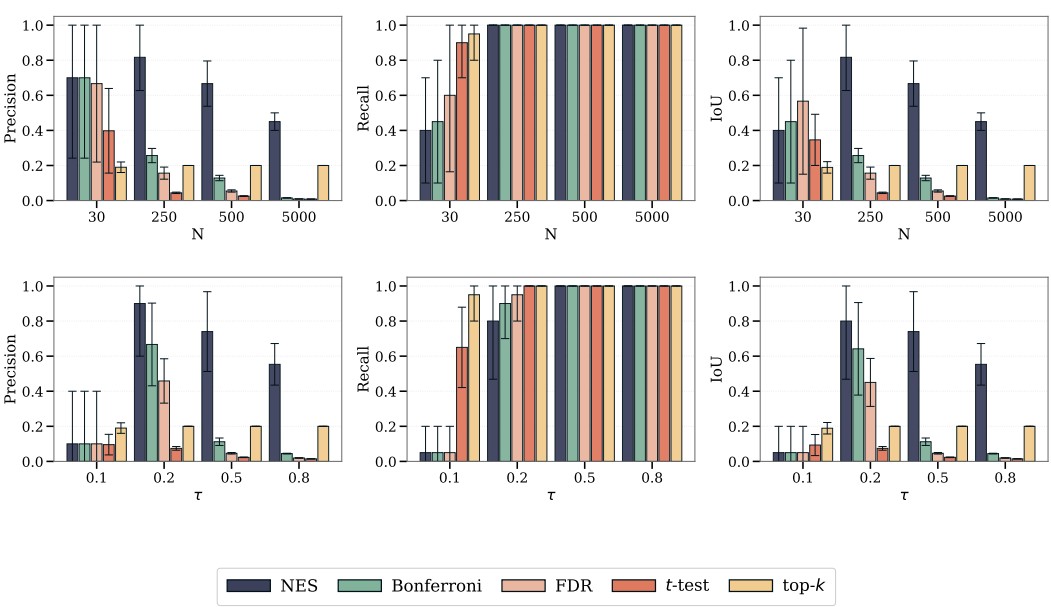

Figure 21: **Ablation SAE non-linearity** (top-10). Precision, Recall, and IoU in effect identification varying sample size $N$ (top) and effect size $\tau$ (bottom), with SAE non-linearity by top-$k$ with $k = 10$. NES consistently replicates the main results using top-$k$ with $k = 20$, still overcoming the paradox of Exploratory Causal Inference.

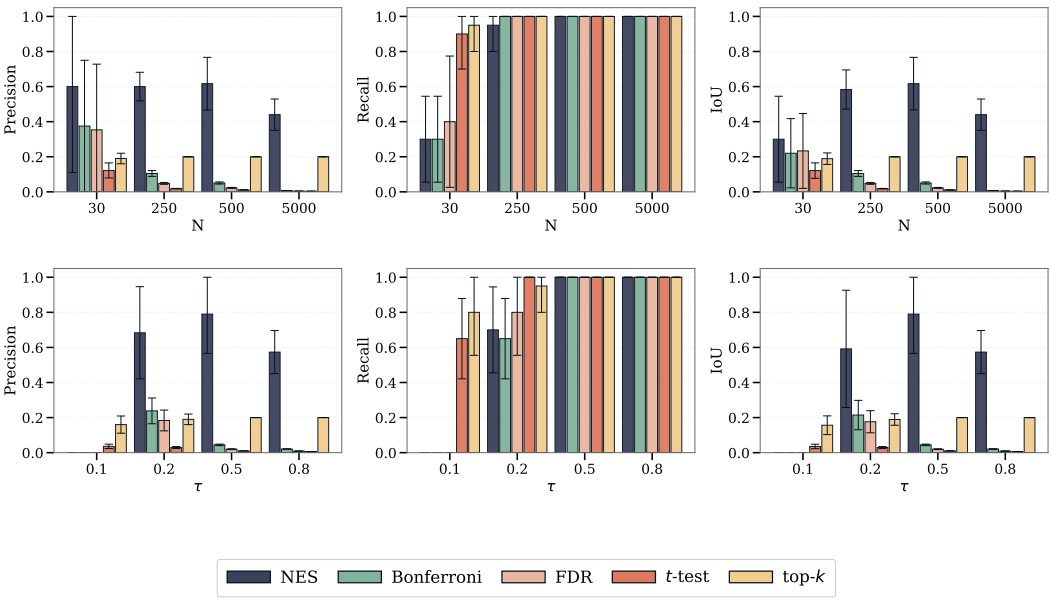

Figure 22: **Ablation SAE non-linearity** (top-50). Precision, Recall, and IoU in effect identification varying sample size $N$ (top) and effect size $\tau$ (bottom), with SAE non-linearity by top-$k$ with $k = 50$. NES consistently replicates the main results using top-$k$ with $k = 20$, still overcoming the paradox of Exploratory Causal Inference.

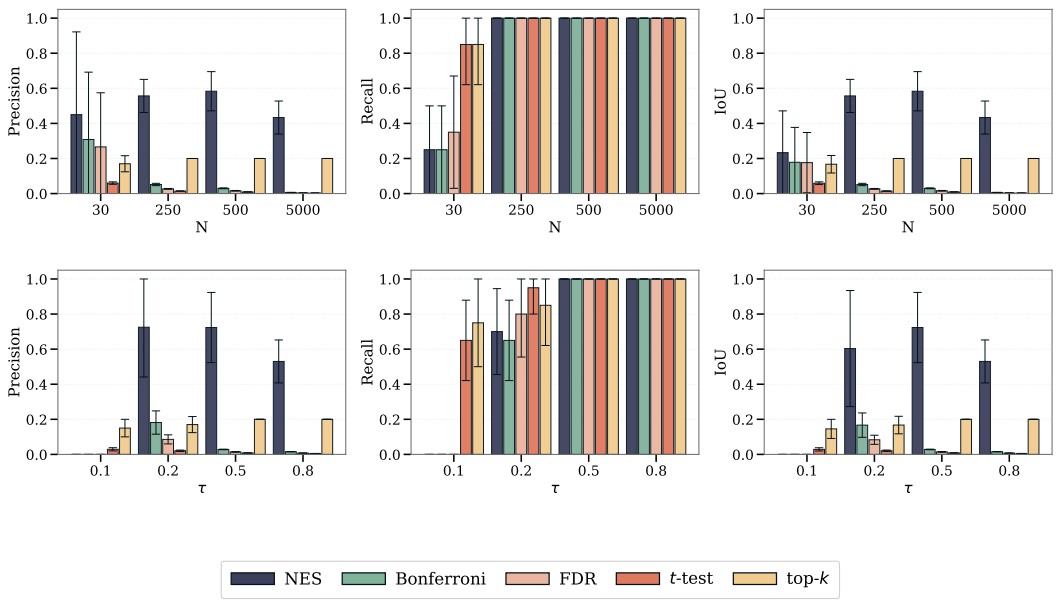

Figure 23: **Ablation SAE non-linearity** (top-100). Precision, Recall, and IoU in effect identification varying sample size $N$ (top) and effect size $\tau$ (bottom), with SAE non-linearity by top-$k$ with $k = 100$. NES consistently replicates the main results using top-$k$ with $k = 20$, still overcoming the paradox of Exploratory Causal Inference.

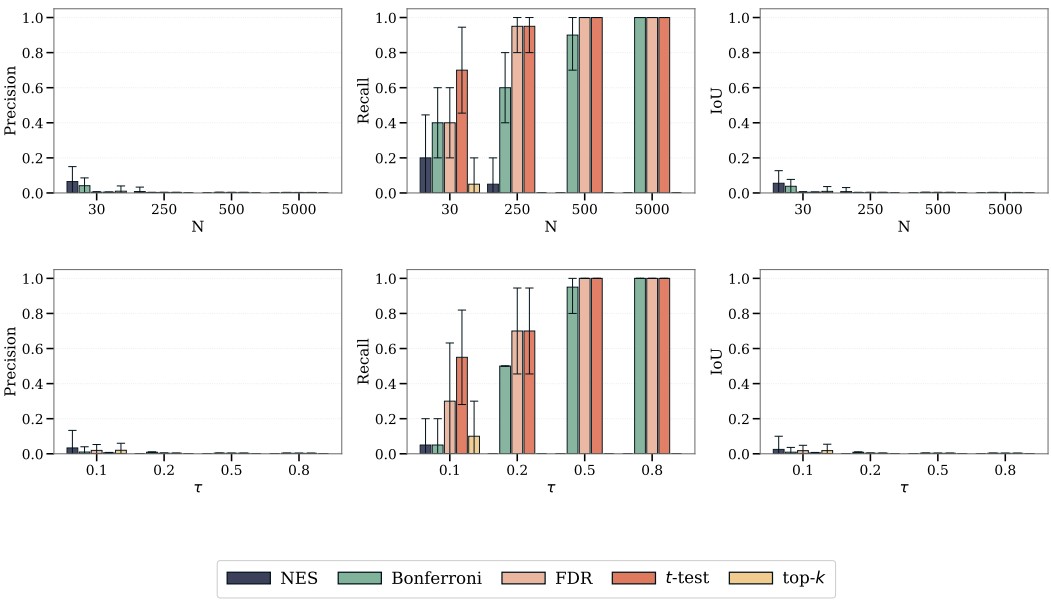

Figure 24: **Ablation SAE non-linearity** (jump-ReLU). Precision, Recall, and IoU in effect identification varying sample size $N$ (top) and effect size $\tau$ (bottom), with SAE non-linearity by jump-ReLU. The principal alignment is violated, invalidating both NES and any evaluation (ill-defined ground truth).

**Seed** Finally, we replicate two more times exactly the same main experiments presented in Section 6), retraining the SAE with a different random seed. Once again, we replicate the results discussed

in Section 6, i.e., NES consistently overcomes the Paradox of Exploratory Causal Inference, while it invalidates all the other baselines. See results in Figures 25-26.

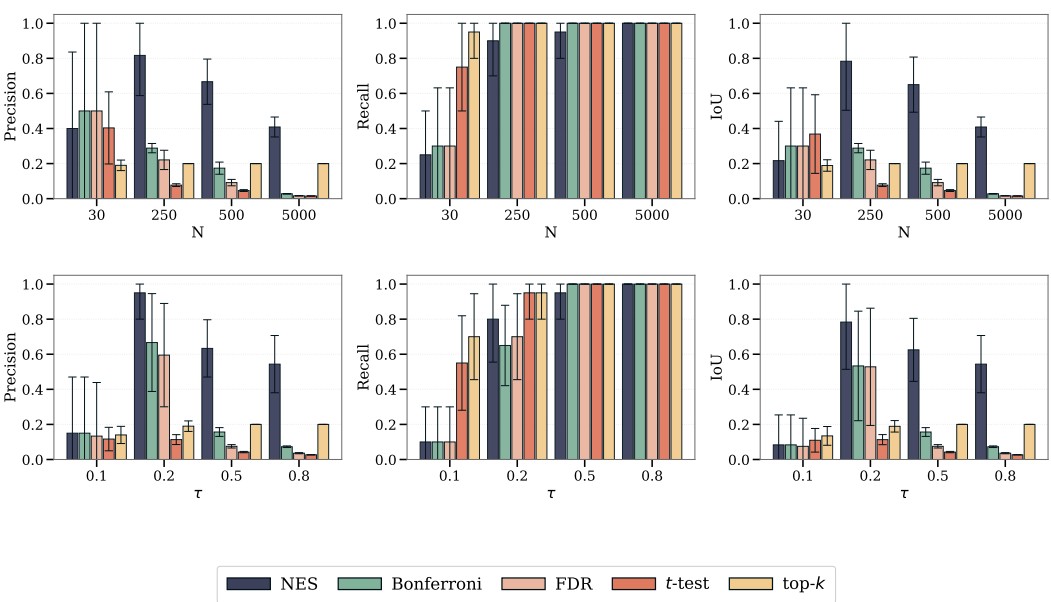

Figure 25: **Replica-1 main results.** Precision, Recall, and IoU in effect identification varying sample size $N$ (top) and effect size $\tau$ (bottom), retraining the SAE with a different random seed (replica 1). NES consistently replicates the main results, still overcoming the paradox of Exploratory Causal Inference.

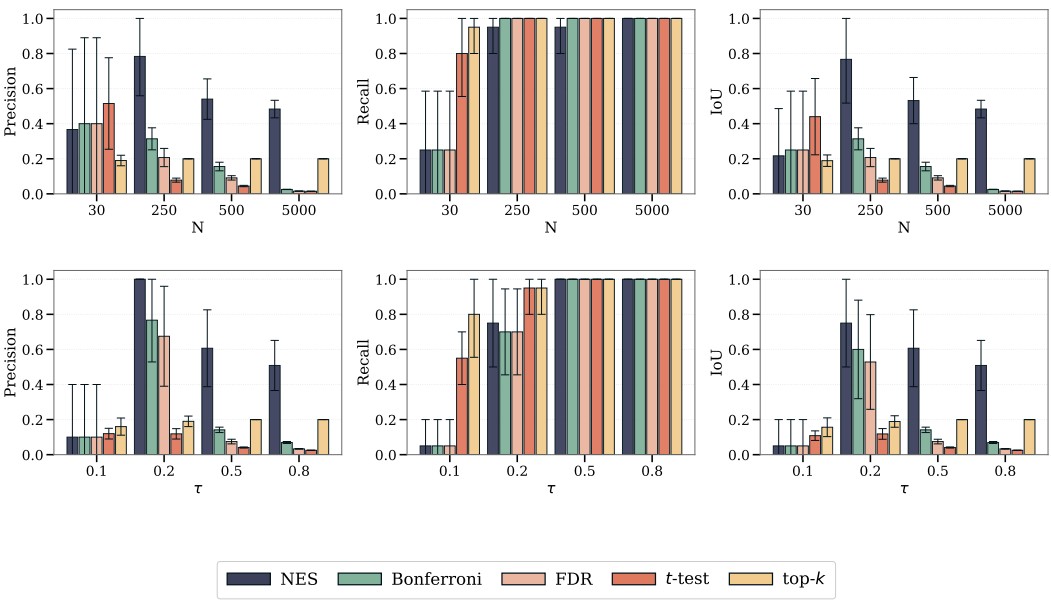

Figure 26: **Replica-2 main results.** Precision, Recall, and IoU in effect identification varying sample size $N$ (top) and effect size $\tau$ (bottom), retraining the SAE with a different random seed (replica 2). NES consistently replicates the main results, still overcoming the paradox of Exploratory Causal Inference.

### E.2.2 VARYING METHOD HYPER-PARAMETERS

We now repeat the main experiments and compare different variants of our methods, varying the multiple-hypothesis testing, using residualization and varying the ATE estimator.

**Hypothesis Testing** We compare three per-round gates inside NES: Bonferroni, FDR, and $t$-test. Same setup as described in Appendix D with extended sample size $N$ and effect size $\tau$ grids, as considered in the additional experiments described in Appendix E.2.3; only the multiplicity rule changes while recursion and residual stratification are unchanged. The results are reported in Figure 27. Bonferroni adjustment and FDR deliver the best recoveries, full recall, keeping Precision $\gtrsim 0$ with sufficient experiment power. *NES-t* is comparable overall, with similar performances with higher power, potentially more exploratory for very small power, i.e., $N = 30$, and less precise in mid-power regimes, i.e., $N \in \{50, ..., 500\}$, $\tau \in \{0.2, ..., 0.6\}$ (weak Paradox of Exploratory Causal Inference). The results suggest generally preferring a multi-hypothesis testing correction, i.e., Bonferroni and FDR, when the power of the experiment is sufficiently high, while considering the $t$-test for a more explorative approach in a very low power regime.

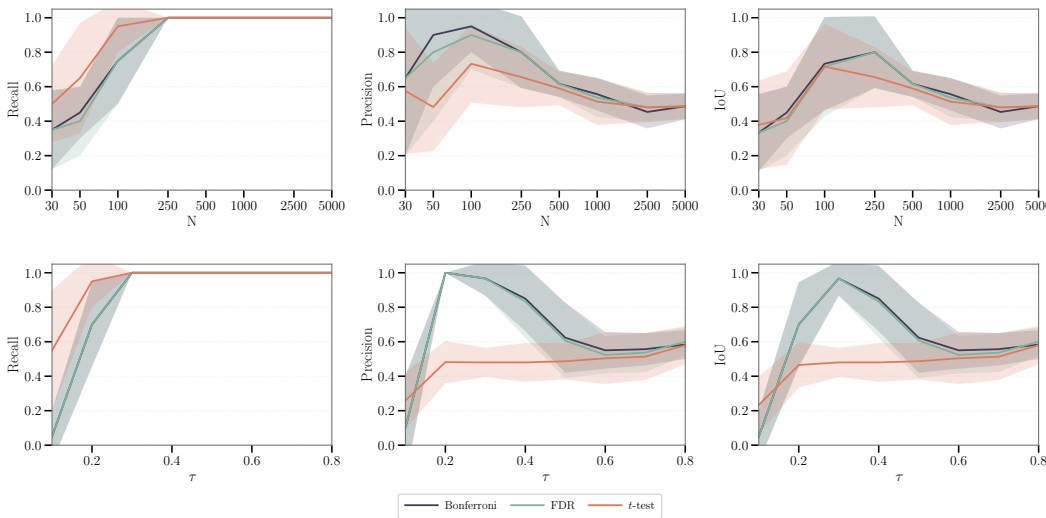

Figure 27: **Ablation multi-hypotheses testing** (within NES). Bonferroni and FDR: best recoveries, full recall, keeping Precision $\gtrsim 0$ with sufficient experiment power; $t$-test: comparable overall, with similar performances with higher power, potentially more exploratory for very small power, and less precise in mid-power regimes.

**Residualization** We then evaluate the benefits in combining the recursive stratification procedure characterizing NES, with or without arm-wise residualization within `NeuralEffectTest` (lines 2-10) for each multi-hypotheses testing method. Same experiment set-up as described in Appendix D. Results are reported in Figure 28. In lower power experiments, i.e., smaller $n$ or $\tau$, arm-wise residualization seems to consistently improve the precision without compromising the recall, for each multi-hypotheses testing variant. In higher power experiments, i.e., greater $n$ or $\tau$, such difference gets negligible as discussed in Theorem 4.1 discussion in Appendix A.

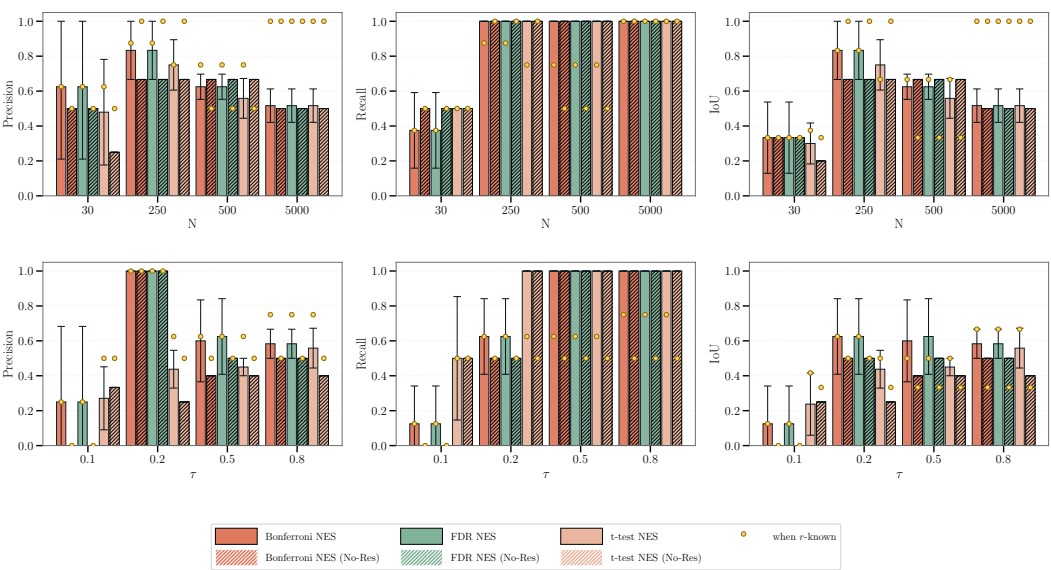

Figure 28: **Ablation arm-wise residualization** (within NET). Additional arm-wise residualization significantly improve the method performances in lower power regime, i.e., smaller sample or reduced effect.

**Average Treatment Effect estimator** Throughout the paper, our per-neuron hypothesis test uses the *associational difference* (AD), i.e., a two-sample $t$-test on the treated–control difference in means. In randomized trials, AD is unbiased for the ATE, but it is not semi-parametrically efficient. A standard variance–reduction alternative is *Augmented Inverse Propensity Weighting* (AIPW; Robins et al., 1994), which orthogonalizes the estimator against misspecification of either the propensity score or the outcome regression. For each code $j$, let $Z_{ij}$ be its activation for unit $i$, $T_i \in \{0, 1\}$ the treatment, and $W_i$ observed exogenous causes. We compute the AIPW pseudo-outcome

$$\tilde{Z}_{ij} = \hat{\mu}_{1j}(W_i) - \hat{\mu}_{0j}(W_i) + \frac{T_i}{\pi(W_i)}\big(Z_{ij} - \hat{\mu}_{1j}(W_i)\big) - \frac{1 - T_i}{1 - \pi(W_i)}\big(Z_{ij} - \hat{\mu}_{0j}(W_i)\big), \quad (36)$$

where $\pi(W) = \Pr(T = 1 \mid W)$, known and constant $\pi = 0.5$ in balanced RCTs, and $\hat{\mu}_{tj}(W) \approx \mathbb{E}[Z_j \mid T = t, W]$ is a nuisance regression. The AIPW estimate of the code-level ATE is $\hat{\tau}_j^{\text{AIPW}} = \frac{1}{n}\sum_i \tilde{Z}_{ij}$; we test $H_0 : \tau_j = 0$ via a one-sample $t$-test on $\{\tilde{Z}_{ij}\}_i$ with robust variance. Figure 29 reports the standalone multiple testing methods performances comparing AD vs. AIPW as neural treatment effect estimator, fixing the effect magnitude $\tau = 0.6$ and across sample sizes $n \in \{30, 500, 5\,000\}$. In our setting, with balanced and randomized treatment assignment ($\pi = 0.5$) and a *single* binary effect-modifier $W$, AIPW yields only marginal efficiency gains: Precision/Recall/IoU curves are essentially overlapping, with small stability improvements for AIPW at the smallest $n$. Crucially, orthogonalization affects variance but *does not* resolve entanglement: the significance–collapse phenomenon for standard multi-testing (Section 3) persists under AIPW, and NES retains its advantage because its benefit comes from recursive stratification (disentangling residual effects), not from how the first-step mean contrast is estimated. Takeaways:

- in pure RCTs with weak, low-dimensional $W$, AD is competitive and simpler,

- AIPW can be preferred when richer exogenous information is available (higher-dimensional $W$, imbalance, or mild protocol deviations), where its variance reduction can translate into earlier detection of the leading effect,

- regardless of AD or AIPW, NES's stratified recursion is the key to avoiding over-discovery under entanglement.

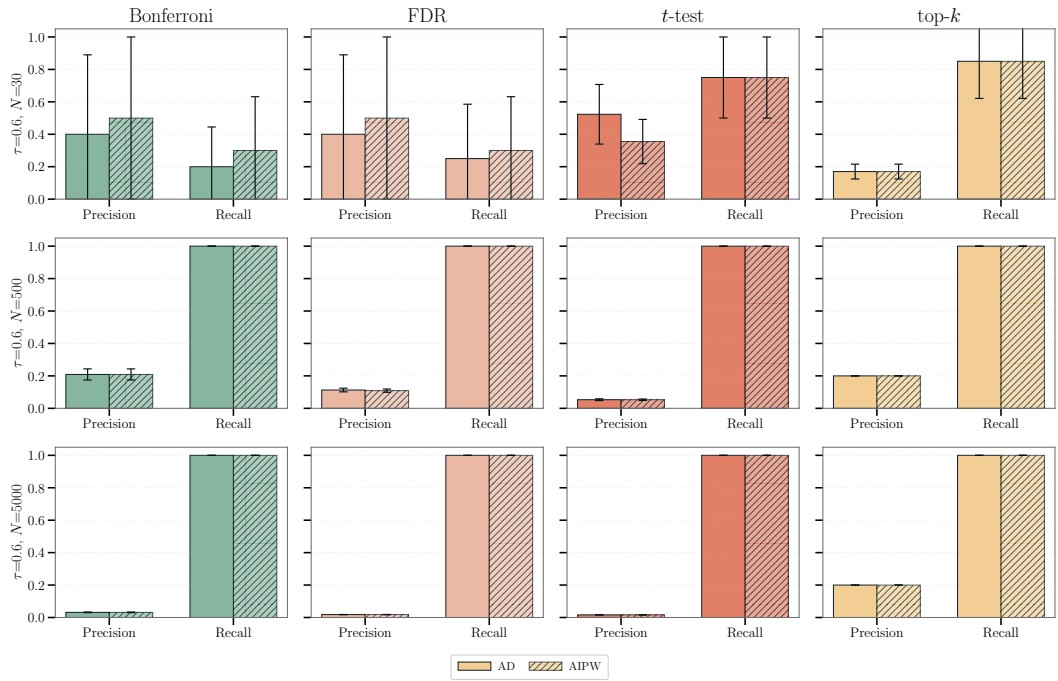

Figure 29: **Ablation Treatment Effect Estimator.** Precision, Recall, and IoU when replacing the per-neuron causal estimator AD (associational difference) with AIPW for vanilla multiple testing criteria (baseline).

### E.2.3    VARYING STRUCTURAL CAUSAL MODEL

**Additional main results**    This paragraph expands the quantitative results illustrated in Figure 5 by reporting all the considered sample sizes and effect magnitudes in the data-generating processes, for two evaluation regimes:

1. *Unknown number of effects* ($r$). Each method returns its *own* set of significant codes. We then report Precision, Recall, and IoU against the ground–truth affected codes.

2. *Known number of effects* ($r$). We assume to know the true number of effects, and we just look at the $r$ highest effects for each method. We again compute Precision, Recall, and IoU (namely, we apply top-2 selection on top of other methods).

As detailed in Appendix D.1, we vary (*i*) the sample size $n \in \{30, 50, 100, 250, 500, 1000\}$ and (*ii*) the ATE magnitude $\tau \in \{0.1, 0.2, \dots, 0.8\}$, holding the semi–synthetic DGP and SAE training protocol fixed. Across both regimes and over the entire grid, NES maintains high Precision and IoU while matching the best Recall of baselines. When the experiment power increases (larger $n$ or $\tau$), vanilla $t$–tests and classical multiplicity corrections (FDR/Bonferroni) exhibit the *significance–collapse* behavior: Recall saturates but Precision drops sharply as leakage neurons become significant, driving IoU toward zero. Enforcing the correct cardinality ($r$ known) mitigates over–selection but does *not* resolve entanglement: baselines still replace a true effect with a leakage surrogate in later picks, keeping Precision $< 0.5$ in the high–power regime. In contrast, NES's residual stratification peels one principal effect component per round and then *stops*, preserving interpretability.

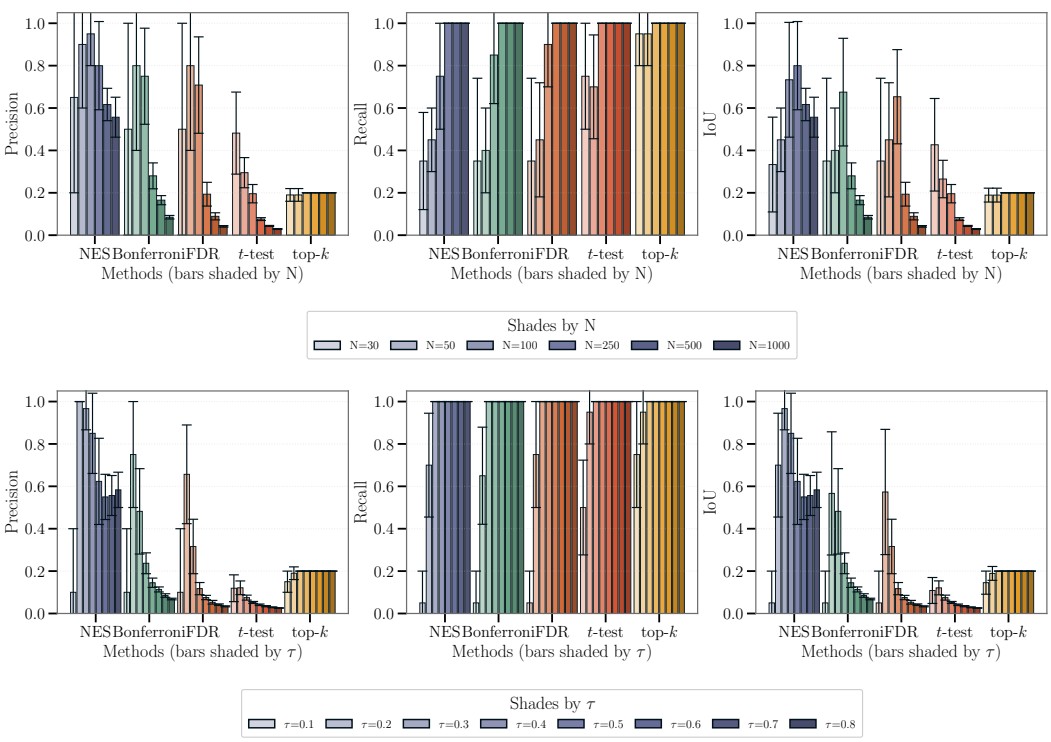

Figure 30: **Full results with $r$ unknown.** Precision, Recall, and IoU for all methods when each returns its own set of significant codes at level $\alpha = 0.05$.

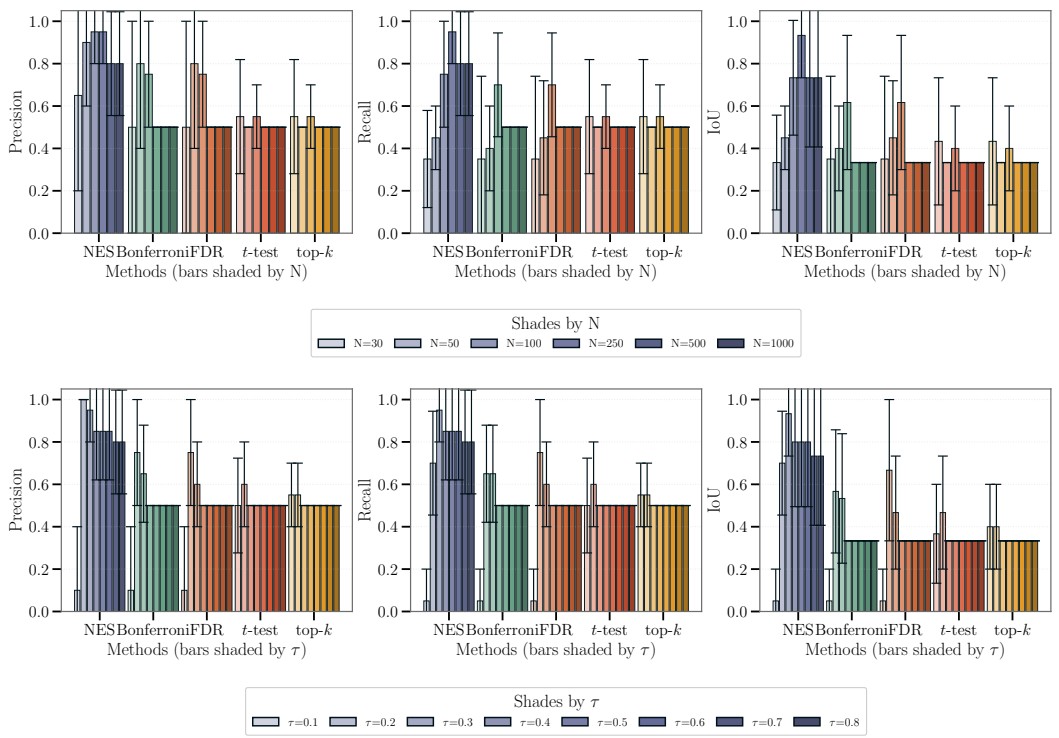

Figure 31: **Full results with $r$ known (top–$r$ selection).** Precision, Recall, and IoU when every method is forced to return the $r$ codes (true number of effects) with the highest absolute treatment effect.

**No effect** We repeat the experiments described in Appendix D, removing both the effects, namely imposing $\tau = 0$. In this setting, a valid discovery procedure should return *no* significant neurons.

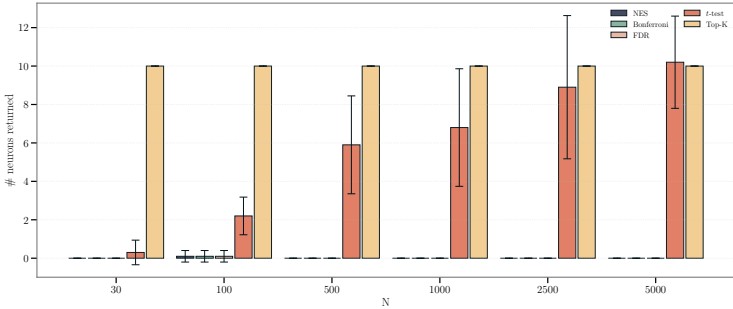

Figure 32: **Zero-effect ablation.** Number of significant neurons retrieved by method when no effects present, i.e., optimal if not retrieving anything.

We keep the data-generating process, foundation model, SAE training, and testing grid over sample sizes $n$ as described in Appendix D, changing only the interventional contrast to $\text{ATE} = 0$. For each method, we record the number of discoveries per run. Across all sample sizes, NES consistently returns an empty set: in the first iteration, no neuron survives Bonferroni at level $\alpha/m$, and the recursion halts. Furthermore, both Bonferroni and FDR also yield essentially zero discoveries. In contrast, the uncorrected $t$-test produces spurious positives (false discoveries), and top-$k$ necessarily reports $k$ indices by design, labeling pure noise as significant. This behavior matches our theoretical intuition: with $\tau = 0$ there is no effect vector to leak into entangled coordinates, so the paradox discussed in Section 3 does not arise; still procedures that control multiplicity (NES via its first-step

Bonferroni gate, Bonferroni, and FDR) appropriately abstain, whereas selection rules that ignore multiplicity (top-$k$, vanilla $t$-tests) over-discover, collapsing the method precision.

**Opposite effects**   To stress–test effect identification under more complex causal structure, we extend the main RCT simulation by allowing the two outcome components $Y_1$ and $Y_2$ to have *different* treatment effects (even complementary). As before, each synthetic unit corresponds to a real CelebA image whose attributes match the realized $(Y_1, Y_2, W)$ triple. We fix $T \sim \text{Bernoulli}(0.5)$ and $W \sim \text{Bernoulli}(0.5)$, and we retain the co–effect model described in Appendix D, but now vary the two ATEs over a grid of ordered pairs $(\tau_1, \tau_2) \in \{(0.2, 0.8), (0.3, 0.7), \dots, (0.8, 0.2)\}$, thus controlling the *gap* between the two causal effects. For each pair, we set the treatment–induced shift for $Y_2$ as

$$p_1^{(Y_2)} = 0.5 + \tfrac{\tau_2}{2}, \qquad p_0^{(Y_2)} = 0.5 - \tfrac{\tau_2}{2}, \tag{37}$$

while $Y_1$ additionally depends on $W$:

$$\Pr(Y_1{=}1 \mid T{=}t, W{=}w) = \begin{cases} p_{11}^{(Y_1)} = 0.5 + \tfrac{\tau_1}{2}, & (t{=}1, w{=}1), \\ p_{10}^{(Y_1)} = 0.2 + \tau_1, & (t{=}1, w{=}0), \\ p_{01}^{(Y_1)} = 0.5 - \tfrac{\tau_1}{2}, & (t{=}0, w{=}1), \\ p_{00}^{(Y_1)} = 0.2, & (t{=}0, w{=}0). \end{cases} \tag{38}$$

This yields a controlled spectrum of settings in which $Y_1$ and $Y_2$ express treatment effects of varying and potentially opposite magnitudes. For each $(\tau_1, \tau_2)$ and sample size $N \in \{500, 5000\}$, we draw $(T, W, Y_1, Y_2)$, attach the matching real image, and apply NES and baselines to evaluate precision, recall, and IoU across the full two–effect grid. The results are reported in Figure 33-34, and consistently replicate the evidence discussed in the main experiments, i.e., NES overcoming the Paradox of Exploratory Causal Inference, as opposed to any other baselines. As expected, with enough experiment power, e.g., $N = 5\,000$, all the methods consistently return the same results for any effects couple, while with lower power, e.g., $N = 500$, both NES and some vanilla baseline can struggle in retrieving all the the effects (perfect recall) if one or both the signals are very limited (as for the main experiments).

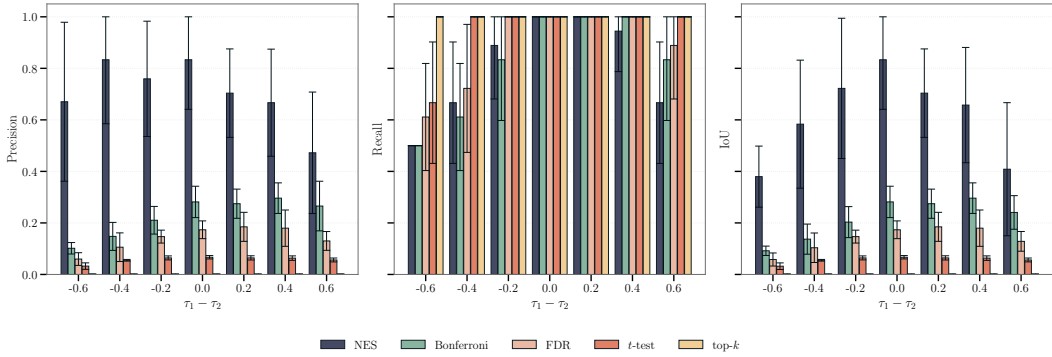

Figure 33: **Ablation opposite effects** ($N = 500$). Precision, Recall, and IoU in effect identification varying the two effects difference in the main structural causal model (fixing $N = 500$). NES consistently replicates the main results, still overcoming the paradox of Exploratory Causal Inference.

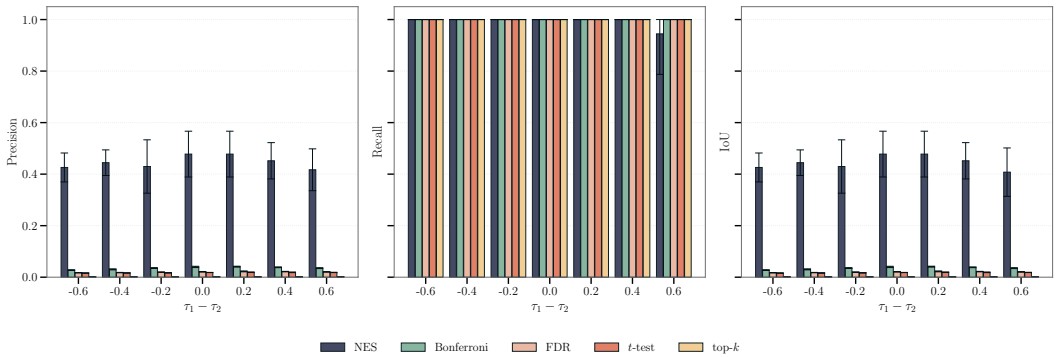

Figure 34: **Ablation opposite effects** ($N = 5\,000$). Precision, Recall, and IoU in effect identification varying the two effects difference in the main structural causal model (fixing $N = 5\,000$). NES consistently replicates the main results, still overcoming the paradox of Exploratory Causal Inference.

**Varying propensity score**  Finally, we repeat the main experiments (fixing treatment effect $\tau = 0.6$ and sample size $N \in \{500, 5\,000\}$) varying the treatment assignment probability $\mathbb{P}(T = 1) \in \{0.2, 0.4, 0.6, 0.8\}$, i.e., propensity score with no dependence on the covariates being a Randomized Controlled Trial. We report the results in Figures 35-35. Once again, both NES and the other baselines replicate the findings described for the main experiments in Section 6. Particularly, NES consistently overcomes the Paradox of Exploratory Causal Inference, which is not the case for the other baselines. The results are consistent and almost invariant with respect to $\mathbb{P}(T = 1)$, even if, as expected, edge values of the propensity score, e.g., 0.2 and 0.8, can slightly affect the method's efficiency.

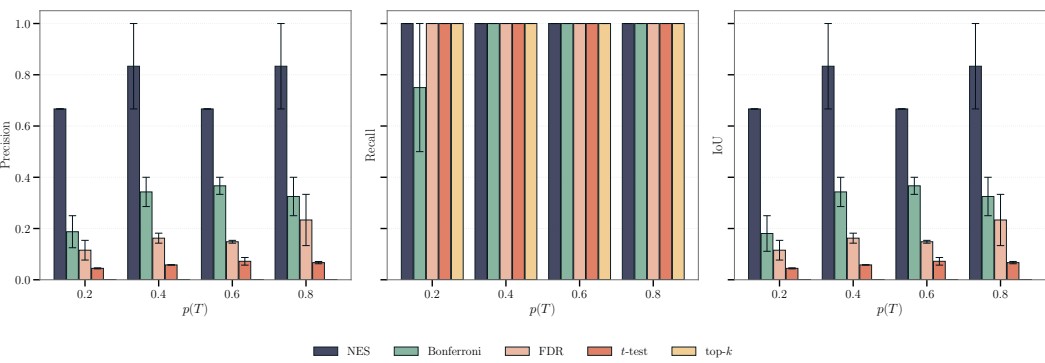

Figure 35: **Ablation unbalanced treatment assignment** ($N = 500$, $\tau = 0.6$). Precision, Recall, and IoU in effect identification varying the treatment assignment probability with sample size $N = 500$. NES consistently replicates the main results in all the regimes, still overcoming the paradox of Exploratory Causal Inference.

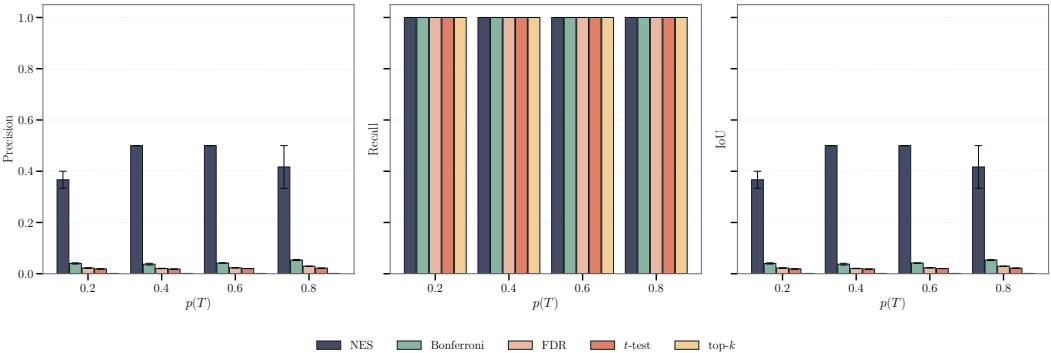

Figure 36: **Ablation unbalanced treatment assignment** ($N = 5\,000$, $\tau = 0.6$). Precision, Recall, and IoU in effect identification varying the treatment assignment probability with sample size $N = 5\,000$. NES consistently replicates the main results in all the regimes, still overcoming the paradox of Exploratory Causal Inference.

### E.3 ADDITIONAL BASELINES

Finally, we further include here two additional non-trivial, still heuristic baselines to compare our method with. Particularly inspired by Causal Feature Learning (Chalupka et al., 2017), which attempts to abstract effects by clustering with respect to $\mathbb{P}(T \mid X = x)$, we propose to consider an additional *neural effect selector* obtained by regressing the treatment $T$ on the SAE representation of $X$, and filtering only the relevant codes. The idea is that the (by design) non-visible treatment information is indirectly retrievable from the effect measurement, and that such information flows through the anti-causal path $Y \leftarrow T$.

**LASSO.** We consider a sparsified logistic regression model (recall that $T$ is binary) with an $\ell_1$ penalty to enforce sparsity on the SAE coefficients. Given SAE codes $Z \in \mathbb{R}^{N \times m}$ and treatment labels $T \in \{0, 1\}^N$, logistic regression models

$$p(T_i = 1 \mid Z_i, \beta) = \sigma\left(\beta_0 + Z_i^\top \beta\right) = \frac{1}{1 + \exp\left(-\beta_0 - \sum_{j=1}^m Z_{ij}\beta_j\right)}, \tag{39}$$

and fits parameters by minimizing the penalized negative log-likelihood

$$\hat{\beta} = \arg\min_{\beta \in \mathbb{R}^m} \left\{ -\sum_{i=1}^N \left[ T_i \log p_i + (1 - T_i) \log(1 - p_i) \right] + \lambda\|\beta\|_1 \right\}, \tag{40}$$

where $\lambda > 0$ controls sparsity. The resulting selected neurons are

$$\mathsf{S}_{\text{lasso}} = \{j \; : \; \hat{\beta}_j \neq 0\}. \tag{41}$$

**Stability-Selected LASSO.** To mitigate instability due to correlated SAE codes, we additionally consider *stability selection*. Given a fixed regularization strength $\lambda$ (e.g., tuned via cross-validation), we repeatedly refit the LASSO model on bootstrapped subsamples of the data. For each bootstrap iteration $b = 1, \dots, B$, let $\hat{\beta}^{(b)}$ be the fitted coefficients. We then define a stability score

$$\Pi_j = \frac{1}{B} \sum_{b=1}^B \mathbb{I}\left(\hat{\beta}_j^{(b)} \neq 0\right), \tag{42}$$

and select those neurons consistently appearing across many subsamples:

$$S_{\text{stable}} = \{j \; : \; \Pi_j \geq s\}, \tag{43}$$

where $s \in (0, 1)$ is a high-confidence threshold. In practice, we consider $s = 0.6$.

We replicate once again the main experimental setup described in Section D, additionally including the two novel baselines, and report the results in Figure 37. Interestingly, the new baselines reach

comparable performance to vanilla multiple-testing methods, despite explicitly exploiting the anti-causal path and never measuring the neural effect directly. With sufficient experimental power, they also reach full-effect recall, but—similarly to vanilla multi-testing—still suffer from the Paradox of Exploratory Causal Inference. Indeed, we are not aware of any theoretical result guaranteeing that these heuristics should recover the standalone ground-truth affected outcomes among all candidate neurons. Stability selection appears to boost vanilla LASSO performance, and with sufficient power, these approaches slightly outperform vanilla multi-testing (although precision remains unsatisfactory, i.e., $\ll 1$).

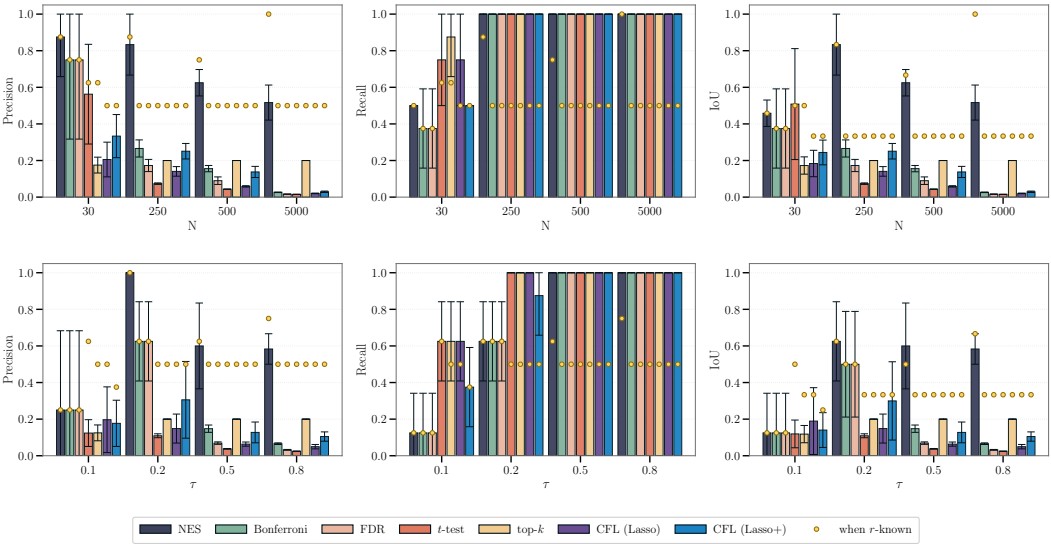

Figure 37: **Additional Baselines from Causal Feature Learning.** Precision, Recall, and IoU in effect identification, including two baselines inspired by effect abstraction in Causal Feature Learning using LASSO with or without stability selection. NES consistently replicates the main results in all the regimes, still overcoming the paradox of Exploratory Causal Inference.

