# OpenReview forum: "Exploratory Causal Inference in SAEnce"
_ICLR.cc/2026/Conference — ICLR 2026 Oral_

### Official Review · Reviewer_waNL · 2025-10-31

**Soundness:** 3
**Presentation:** 3
**Contribution:** 3
**Rating:** 8
**Confidence:** 4

**Summary:**

The paper proposes Exploratory Causal Inference. Starting from raw, high-dimensional measurements $X$ collected in randomized trials where the specific outcome variables $Y$ are unknown, the authors extract representations with a foundation model (FM), reparameterize them with a sparse autoencoder (SAE), and then test for treatment effects at the level of SAE codes. They identify a core failure mode, the paradox of exploratory causal inference, where classical multiple testing (even with Bonferroni/FDR) flags all entangled neurons as “significant” as sample size or effect size grows, destroying interpretability. To address this, they introduce Neural Effect Search (NES), a recursive, arm-wise residual stratification procedure that peels away effects one by one and (under assumptions) consistently recovers the true effect subspace. They present semi-synthetic experiments (CelebA-based) and a real randomized ecology trial (ISTAnt ants) where NES identifies a behavior (grooming) and a nuisance/background correlate.

**Strengths:**

Originality. The manuscript frames an under-explored empiricist alternative to “prediction-powered” causal inference by discovering what was affected (unknown $Y$) rather than who is affected (heterogeneity over $W$). The formal paradox and its asymptotics represent a crisp conceptual advance that explains why usual multi-testing fails under entanglement.

Quality. The NES procedure is simple, causally motivated (conditioning/stratification to remove leakage and collider bias), and backed by a consistency theorem under a clear SCM. The proofs (appendix) show how arm-wise residualization preserves the remaining causal contrasts.

Clarity. Pipeline and problem setting are clearly contrasted with classical RCTs and “prediction-powered” approaches. Figures and toy examples clearly communicate the paradox and the proposed fix.

Significance. If validated more broadly, NES could become a practical hypothesis-generation and pruning layer for large, instrumented experiments (biology, ecology, imaging, etc.), reducing annotation burden while keeping statistical guardrails. The empirical ecology case shows it can recover a known behavior (grooming) and also surface a background artifact that informs experimental redesign.

**Weaknesses:**

Identification hinges on SAE/FM assumptions. Consistency requires that the true effect directions live (approximately linearly) in the SAE code space and that the SAE reaches a sufficiently mono-semantic regime. This is acknowledged as the biggest limitation in the manuscript. The paper would benefit from stress tests across different FM/SAE choices, sparsity regimes, and code dimensionalities, plus diagnostics for “are we in a regime where NES is trustworthy?” (e.g., leakage indices before/after NES and stability under code re-initialization).

NES uses Bonferroni within rounds, but the recursion adaptively selects and conditions on previous picks. The theorem covers recovery of directions under asymptotics. However, finite-sample error rates (FWER/FDR) across the entire procedure remain unclear. The authors are suggested to quantify family-wise error or provide a selective-inference style bound for the full recursion (or a practical stopping rule tied to post-selection adjusted p-values).

The toy setups are helpful but narrow. The authors may consider adding settings with (i) multiple effects with very different magnitudes; (ii) nonlinear superposition in codes; (iii) batch effects and arm imbalance; (iv) mis-specified FMs (domain shift). It would be helpful to report precision/recall/IoU when ground-truth effects are weak and overlapping, and ablate stratification choices.

The real trial yields two neurons; one aligns with grooming ($F1\approx0.40$), another with palette background correlated with treatment due to small $n$. This convincingly demonstrates hypothesis surfacing, but decisions based on neural codes remain risky. Strengthen the case with expert-verified labeling for the surfaced codes (beyond top clips), pre-registered confirmatory tests, or a hold-out colony/day to show replicability.

**Questions:**

1.	Can you characterize the overall FWER/FDR of the entire recursive NES (not just within-round Bonferroni)? Any feasible post-selection correction or stability-selection variant?
2.	What quantitative leakage/entanglement metrics do you compute pre- and post-NES? Can you report these alongside effect discoveries to guide practitioners? E.g., your leakage set/index definition as a reported diagnostic.
3.	How stable are discovered effects across different FMs (e.g., DINOv2 vs. CLIP/SigLIP), code sizes, sparsity penalties, and random seeds? Any consensus-NES idea where you intersect effects across runs?
4.	In the ecology study, could you provide bootstrap confidence intervals for NES effect sizes on residualized codes, or permutation tests that respect the randomization?
5.	Beyond “stop when no rejections,” is there any data-dependent stopping that protects against over-peeling? And for multiple effects, how robust is the ordering (largest first) under moderate entanglement?
6.	Could NES results predict downstream manual labels or behavioral rates on an unseen experimental batch, thereby validating discovered effects?

---

> ### Author Response · Authors · 2025-11-21
> **New extensive ablations and theory clarification**
>
> We thank reviewer @waNL for the detailed and constructive feedback. We answer here to both the experimental and theoretical concerns raised. Please refer to the 'General answer' for all other modifications we have made to the draft (new experiments, clearer theory, improved writing).
>
> **Experiments**
>
> - **Explicit assumption testing (Q2)**:
>
>   The treatment effect entanglement in the neural representation is generally untestable without supervision on the (ground truth) affected outcomes. Nevertheless, we improved the exposition of the required assumptions (see Appendix A.3) and now added ablations on their validity and violation, assuming known ground-truth for testing, i.e., outcomes annotations (see Appendix E.1). In practice, we envision practitioners to simply rely on models that are known to work well with linear probes on similar data (which may also be tested with observational data), and SAE implementations that they can interpret. We’d like to remark once more that our method should not replace rationalist approaches at this point, but complements them in exploratory settings. As such, the explicit scientific interpretation is crucial.
>
> - **Extensive method consistency ablation (Q3)**:
>
>   We then embrace the reviewer's concern about stress testing our method's consistency, and accordingly, we added extensive novel ablations on both method settings and more challenging problem instances, i.e., data-generating processes. All the additional experiments are now reorganized in Appendix E.
>   In particular, we varied:
>   - the measurements encoder (foundational model, SAE dimension, SAE non-linearity, and training randomization),
>   - the method hyperparameters (hypothesis testing and causal estimator),
>   - and the data-generating processes (full grid of main experiments, no effect, opposite effects, varying propensity score).
>
>   According to our theory, our method consistently overcomes the Paradox of Exploratory Causal Inference in all the settings differently from all the other baselines. The unique exception is using SAE with jumpReLU non-linearity due to the principal alignment assumption strong violation (tested in Appendix E.1).
>
>
> **Theory**
>
> - **False discovery rate O($n^{-1/2}$) convergence (Q1,Q5)**:
>
>   Characterizing finite-sample FWER/FDR of the full recursive NES is nontrivial (beyond the scope of this paper) because the procedure is adaptive: each round tests data-dependent hypotheses, so standard fixed-design multiple-testing guarantees do not apply. However, Theorem 4.1 already implies asymptotic global error control: both the overall FWER and FDR of the entire NES recursion converge to 0 as $n \rightarrow \infty$, under the required assumptions. We now state this explicitly in the appendix. The explicit $\sqrt{n}$ separation rate further clarifies how global FWER/FDR vanish asymptotically, i.e., $O(n^{-1/2})$, as in standard CLT-based tests. According to that, there is no motivation to adapt the built-in stopping criteria within NES (Q5). However, combining the results from different runs, e.g., by stability selection, could instead practically boost the method's robustness and reduce uncertainty (in all our synthetic experiments reported by repeating the analyses 10 times, resampling the experimental data). Preliminary evidence of advantages in aggregating the results from different runs is shown for the novel baseline we introduced in Appendix E.3. These practical extensions of NES are beyond the scope of this paper, and we hope they will be explored in future work together with more extensive and interdisciplinary benchmarking.
>
> - **ECI consistency is not sufficient for PPCI (Q4, Q6)**:
>
>   We propose NES as a method for effect identification, not inference, i.e., effect estimation with confidence intervals. Indeed, its extension to valid Prediction-Powered Causal Inference is not straightforward for two reasons:
>   - without supervision, prediction uncertainty and potential biases are not measurable,
>   - neuron interpretation may not be unique, and different hypotheses could be considered.
>
>   Minimal supervision, ignored in our setting, has to be considered for any valid causal effect estimation from a standalone Exploratory Causal Inference pipeline. As stressed at the end of Section 2, “*the empiricist view should not replace the rationalist one, but enrich it with additional data-driven hypotheses*”.

---

> > ### Author Response · Authors · 2025-11-27
> >
> > We kindly ask the reviewer if their questions were answered to their satisfaction and remain available for any further discussion.

---

### Official Review · Reviewer_rdaU · 2025-11-01

**Soundness:** 3
**Presentation:** 2
**Contribution:** 4
**Rating:** 8
**Confidence:** 2

**Summary:**

The authors focus on an empiricist, rather than rationalist, approach to analyzing data from RCTs.  The increase in large data collections opens up a increasing range of datasets where interventions were performed without clear hypotheses or outcome targets.  In such domains, it can be useful to explore the space of possible outcomes for a given treatment, but the high dimensionality of the observation space can make disentangling concrete outcomes challenging and can lead to the detection of large numbers of spurious correlations if care isn't taken.  To tackle these challenges, the authors propose a new procedure that extracts a representation from the input data using a foundation model, applies a sparse autoencoder to separate out atomic outcomes.  However, these dimensions may still be strongly entangled.  The authors account for this with an algorithm they propose, NES, that recursively identifies and removes outcomes from the SAE representation, resulting in a set of possible outcomes that can be interpreted by an analyst.  The authors explain how this procedure can be implemented and how it avoids issues around multiple hypothesis testing, and evaluate its performance with semi-synthetic and empirical datasets.

**Strengths:**

Overall, I think the this is a great paper.  The problem definition, though hard to understand initially, is novel and compelling.  The authors make a solid case about the value of empirical exploratory causal inference, and their approach seems well-reasoned.  Apart from the issue of the problem statement needing more upfront description/examples, the rest of the paper is well-presented, and the writing is clear and easy to follow.  For an approach to solving a problem that hasn't been studied much before, I think the authors did a good job at still doing a reasonable evaluation.  I also appreciate the authors' scientific rigor with respect to the dangers of multiple hypothesis testing and the need for a human analyst.

**Weaknesses:**

While I generally think the central idea of this paper is strong, the presentation is the weakest part.  The problem definition the authors are working with (RCTs without a directly measured, or even defined, outcome) is interesting but very uncommon in the literature, and the authors' description of it in the paper is insufficient.  The setting seems to be the same as that used in Cadei et al. 2024 and 2025, and it was only through reading the examples in those papers that I was able to really understand the problem domain.

The first part that this lack of sufficient explanation hit for me was at the top of page 2, where the authors state that "Clearly, [understanding the spread of disease from fine-grained social interactions] can be dramatically accelerated with computer vision, using the predictions of a model as input for causal inference pipelines (Cadei et al, 2025)."  Without following the reference, this is the first time "computer vision" or images are brought up, so the "clearly" definitely doesn't work for the reader here.  Similarly, the following sentence about deciding what to annotate is equally unclear when the authors haven't yet given us the framing of the problem as one where outcomes must be inferred from pixels in images.  The authors then, in line 64, mention that the problem they consider is RCTs where the effect is measured indirectly, possibly through imaging.  However, the concept of an RCT without an explicitly measured outcome is counterintuitive in most domains.  When we eventually get details about the ant grooming example, it makes sense and is a compelling problem definition, but until then, the problem statement remains incredibly murky.

My recommendation would be to put the ant grooming example in the introduction.  Even with the brief discussion of the ant grooming problem in this paper (first around like 146, and then again in the experiment section), the details are very light, and it wasn't until I read the first two sections of Cadei et al. 2024 that that data set (and thus your problem definition ) really became clear to me.  I think the main piece that was missing from this paper is why outcomes wouldn't be directly observable in an RCT.  Giving a concrete example and explaining that an outcome could be, for example, a specific type of ant grooming behavior (which can only be inferred by an expert watching the frames of a video) would make the paper much easier to follow.  And then you can easily explain that, while in some cases experts may have specific behavior they're expecting to change upon treatment, this may cause them to miss other interesting behaviors that are also inferable from the video.  Through the rest of the paper, there are other parts that seem like they would benefit from example, so having a running example set up early on would help a lot. (e.g., on page 3, when discussing Figure 2 center and right)

The experiments are interesting and well-done.  However, I wish the authors had actually described what the "palette background" effect was in the ISTAnt experiment.  Even just putting it in the appendix would help.  As-is, needing to go to another paper to even understand the empirical evaluation isn't ideal.  I'm not really even sure what the description on line 430 means ("top right black color mark in the top left position in the first 4 batches of videos") - in the bottom row of Figure 5, I do see black marks in the top right of the first two images, but I'm not sure what "the top left position" means there.

I guess this doesn't really diminish the contribution, but it's very odd to have what looks like an algorithm name in the title of your paper (SAEnce) but then not actually have that be the algorithm name (NES).  From Figure 1, I assumed that NES was just part of an overall procedure called SAEnce, but then you never actually use the term SAEnce outside the title.

While not rampant, there are some grammatical issues that could be cleaned up.  These didn't affect my score, but just for your information, here are some that I noted early on:
- line 43: "or more in general" -> "or more **generally**"
- line 46: "modern science started embracing the creation of atlases" -> "modern science **has started embracing the creation of atlases**
- line 48: "(Weinstein et al., 2013), imaging of cells ..." -> "(Weinstein et al., 2013), **and** imaging of cells ..."
- line 72: "One effect after the other" doesn't fit at the end of that sentence.  You at least need a comma before it (and really, I think it would sound better with a verb, but I'm not quite sure what would fit best - maybe "**, extracting** one effect after the other"?)
- line 127: I think "i.e., T has no causes" should be in parentheses

**Questions:**

On line 143, what does it mean to "assume T is not directly visible in X".  For example, thinking of the ant grooming example where the treatment is some substance that the ants were given, does this just mean that none of the substance is directly seen in any of the pixels of the images?  If so, how does this map to a "double-blind randomized trial"? (if T not being directly visible in X just means something like not seeing the substance pixels in the image, that doesn't seem to relate to whether or not the ants taking the substance knew that they took it or not)

When discussing Figure 2 (center) in the middle of page 3, this seems to be strongly related to work by Cadei et al., 2024 and 2025.  However, in that 2024 paper, there's a similar, but different, causal structure (Figure 1 in that paper) that seems to describe the same problem where we only observe a high-dimensional observation view of the outcome of interest.  Rather than X being caused by W and Y (as in Figure 2 (center)), they have X being caused by T and W and then causing Y.  Can you explain the reason for this difference, despite the problem settings, at least superficially, seeming very similar?

On line 158, when talking about Figure 2 (right), you state that you have T being independent of W given Y.  However, that doesn't appear to be true in Figure 2 (right) - it looks like conditioning on Y, which is a collider between T and W, would induce dependence.

I feel like I'm missing something about the title.  I get that your approach uses SAEs, and it looks like some sort of play on "seance", but "SAEnce" doesn't appear to be the name of your algorithm (which is NES), and you never refer to the term "SAEnce" in the paper.

---

> ### Author Response · Authors · 2025-11-21
>
> We thank the reviewer @rdaU for the very detailed and valuable feedback. We first reply to the major concerns and then answer the remaining questions individually. Please refer to the 'General Answer' for all other modifications we have made to the draft (new experiments, clearer theory, improved writing).
>
> **Problem Formulation is novel but hard to understand**
>
> - We recognise the presentation weaknesses raised by the reviewer, especially for the novel and unexplored problem setting considered. Accordingly, we strengthened the practical examples in the introduction, starting in line 42, and added a box explicitly describing the setting in the application considered in the experiments. We further revisited the first two sections' phrasings accordingly, hoping overall to make the problem understanding more accessible.
>
> **Causal Model: difference from Cadei et al. (2024), and Causal Abstraction interpretation**
>
> - We indeed reconsider the experimental data introduced by Cadei et al. (2024), but offering a different causal interpretation of the data-generating process: while they define the behaviors as a posteriori interpretation of the video recordings, we assume the existence of some latent generating factors for the recordings, e.g., ants' behaviours. Our assumption reflects the established representation learning setting with a latent data-generative process, while their model reflects the scientist's labeling point of view, ignoring any latent data-generative process. However, without the latter, it makes almost no sense to talk about treatment effect abstraction to a posteriori concept definition. In their paper, instead, either view makes no practical difference (corresponds to the old debate in representation learning, whether the labels cause the images or the images cause the labels).
> - The reviewer pointed out a mistake in our explanation of the causal abstractions with conditional independence. We have now fixed it. We’d like to remark that the conditional independence we wrote is never used in the method, and we had written it purely for intuition purposes. Thank you for spotting this, and we apologize for the mistake.
>
> **Other discussions:**
>
> - *Additional results interpretation*:
> We appreciate the detailed interest in the quantitative interpretation of the second neural effect in the ISTAnt experiment, i.e., a specific black-pen marking on the background for the dish-palettes positioning. We extend and add its quantitative interpretation in Section 6.2, and further stress NES as testing for correct experimental design, easily interpretable by domain experts.
>
> - *Unobservable Treatment Assignment*:
> Assuming unobservable treatment assignment means that the treatment assignment is not directly distinguishable in the measurement without modeling its effects, and it is the gold standard of experimentation (double-blind randomized trials). In our case, it means that both the control and treatment interventions have similar or no appearance, so that the neural network has to model the effect to distinguish the trial’s arms, e.g., on ISTAnt, both the control and treatment interventions consist of a visually indistinguishable solution drop. While the blindness of the annotator is necessary for successful Exploratory Causal Inference, the blindness of the individual with respect to the treatment assignment is important regardless, for the identification of the causal effect itself (i.e., no placebo effect).
>
> - *Phrasing and Title*:
> We appreciate all the detailed phrasing comments we have already integrated. Concerning the paper naming, we agree NES is the main method. However, both Science and SAE representation are the object of interest for Exploratory Causal Inference, and we proposed to combine them in the title with a pun (Science → SAEnce). Do you believe we should explicitly stress its meaning in the introduction or conclusion? We thought it was self-explanatory.

---

> > ### Author Response · Authors · 2025-11-27
> >
> > We kindly ask the reviewer if their questions were answered to their satisfaction and remain available for any further discussion.

---

### Official Review · Reviewer_ZiiZ · 2025-11-01

**Soundness:** 3
**Presentation:** 3
**Contribution:** 3
**Rating:** 8
**Confidence:** 4

**Summary:**

The paper proposes an exploratory causal inference pipeline for RCTs with unstructured/high-dimensional outcomes. The main contribution is Neural Effect Search and its application on real experimental ecology trial.

**Strengths:**

The paper is very well-written with clear problem framing and clear methodology paired with real-world case study. It is a promising work for exploratory science workflows.

**Weaknesses:**

Relation to variable selection. Since NES aims to identify which SAE codes carry treatment effects, could you compare it against classical variable-selection baselines for effect modifiers? E.g., Lasso with group/hierarchical penalties, knockoffs or stability selection for support recovery. Such comparison may clarify what NES adds beyond standard selection.

**Questions:**

**Q: Computing** v_k **when** Y **is unobserved.**

In eq.4 you define the “neuron effect” vector for factor Y_k as $v_k := \mu(Y_k{=}1)-\mu(Y_k{=}0)$, which you later use for NES. However, in your main RCT setting the semantic factors Y=(Y_1,\dots,Y_r) are not observed. Could you clarify how v_k is computed in practice?

---

> ### Author Response · Authors · 2025-11-21
>
> We thank the reviewer @ZiiZ for the valuable feedback. We answer here to the two concerns raised. Please refer to the 'General answer' for all other modifications we have made to the draft (new experiments, clearer theory, improved writing).
>
> **New Baselines**
>
> We appreciate the insightful request to ‘*test our method against classical variable-selection baselines for effect modifiers*’ to clarify ‘*what NES adds beyond standard selection*’. Such methods cannot directly be implemented in our setting, having no supervision on the effect. Nevertheless, we extended our analyses with additional baselines using (i) LASSO and (ii) LASSO with stability selection (as suggested) for the anticausal problem of modeling the treatment assignment probability. Indeed, following the Causal Feature Learning principle for effect abstraction by clustering P(T|X=x), we expect that all the codes in X representations bringing information about the treatment T are doing it because of the anticausal path X←Y←T (assuming treatment T is directly unobservable in the measurement X by design). Practically, we model such a relationship via a Logistic Regression with L1 regularization, i.e., LASSO, with or without Stability Selection. Despite promising and potentially overperforming vanilla multiple testing (without direct neural effect measurements), these baselines still suffer the Paradox of Exploratory Causal Inference (good recall but dropping precision). Full details about these additional baselines are reported in Appendix E.3.
>
>
> **Neural Effects Representation Clarification**
>
> We further embrace the reviewer’s request in clarifying the role of the neural outcome effect vector for a factor $Y_k$. It is indeed just an auxiliary definition; we don’t compute or test directly (differently from the neural treatment effects), and we introduce it only to define the effect entanglement, i.e., leakage set, and the neural treatment effect linear decomposition (used in NES consistency proof). To simplify, we rename it as “factor $Y_k$ neural representation”, we remove the redundant code-mean maps definition, and specify its auxiliary objective in the discussion. See Section 3 for the corresponding changes.

---

> > ### Author Response · Authors · 2025-11-27
> >
> > We kindly ask the reviewer if their questions were answered to their satisfaction and remain available for any further discussion.

---

### Official Review · Reviewer_CiiA · 2025-11-05

**Soundness:** 3
**Presentation:** 2
**Contribution:** 2
**Rating:** 4
**Confidence:** 3

**Summary:**

The paper aims to discover treatment effects on targets that are unknown. As opposed to prediction-powered causal inference, in which the goal is to perform experiments to study hypotheses on known targets, the paper focuses on exploratory causal inference, where the targets are discovered from data, and causal effects are interpreted afterwards. The proposed approach first obtains representations of measurements through a pretrained foundation model and then reparameterizes them using a sparse autoencoder. Results are provided that state that the resulting variables are entangled, and they are processed through a procedure called Neural Effect Search to produce a set of disentangled variables.

**Strengths:**

1. A method of performing experimental studies that is not inherently biased by a predetermined hypothesis can be quite useful. The idea of learning targets instead of having a preestablished one is a novel concept.

2. Experiments provide some interesting insight into the consequences of the proposed approach.

**Weaknesses:**

3. The problem does not seem very well-defined from the perspective of variable definitions. Notably, it is unclear what makes something an outcome (part of $Y$) or a measurement (part of $X$). Especially if in this problem setting, $Y$ is constructed by processing $X$ deterministically, statistical independence tests (e.g., to check if $T \perp X \mid Y$) are potentially invalid due to lack of positivity.

4. Guarantees of the proposed approach are mostly leveraging mutual information between variables, but mutual information may not capture causal nuances between variables.

5. The paper could be improved in clarity. Notably, it might help to include a figure of the whole pipeline or perhaps an example.

**Questions:**

6. Is the proposed method restricted to experimental settings in which $T$ is being intervened? Or could it also be applicable in general cases in observational studies such as where $T$ is interpreted as an instrumental variable?

7. Why is it necessary to remap $X$ to a representation from the foundation model?

8. What makes the idea of learning the outcomes different from papers that study causal disentangled representation learning? Is it the idea that outcomes are different from other variables? Why so?

---

> ### Author Response · Authors · 2025-11-21
> **Possible objective misunderstanding**
>
> We thank the reviewer @CiiA for the feedback. We first reply to the major concerns and then answer the remaining questions individually. Please refer to the 'General Answer' for all other modifications we have made to the draft (new experiments, clearer theory, improved writing).
>
> **Neural Effect Search identifies treatment effect by selection, not variable manipulation**
>
> From the reviewer’s summary: “[...] *variables are entangled, and they are processed through a procedure called Neural Effect Search to produce a set of disentangled variables*”. Similarly, the reviewer asked about representation disentanglement (Q8).
>
> Despite partially appreciating the novelty and potential of the work, there seems to be a misunderstanding of our method’s (NES) objective. Our method deals directly with principally aligned but entangled representations and disentangles the effect of the treatment on the neural representation, not the representation itself. This is the crux of our idea, offering a novel link between mechanistic interpretability and causality that pivots from requiring unrealistic representation desiderata. We then enforce neural effect ‘*disentanglement*’ during estimation by recursive (causal) stratification, and not leveraging mutual information directly (W4).
>
> **Problem Formulation clarification**
>
> We appreciate the request for problem formulation clarification (W3) and extending the motivating example (W5). In agreement with reviewer @rdaU suggestion, we revisited the first two sections accordingly, including an explicit motivating example of our setting directly in the introduction, hoping to further clarify the set-up interpretation. According to them, we hope to now clarify that (treatment assignment) positivity for statistical independence tests is not violated by design in our settings, due to the randomized treatment assignment assumption, i.e., experiment.
>
> **Other discussions:**
>
> - *Possible extension to non-experimental data (Q6)*:
> Our method could, in principle, be generalized to the instrumental variable (IV) setting and other settings assuming treatment effect identifiability. However, each setting is characterized by peculiar challenges, e.g., in the IV setting, if the treatment assignment is visible, NES identifies first the most predictive neuron for the treatment and then loses any downstream effect identification by stratifying on it. We don’t believe any of these extensions are generally straightforward without non-trivial supervision to correct the representation or effect estimation.
>
> - *Sparse representation requirement (Q7)*:
> The remapping from raw observations to a foundational model representation, and then the sparse autoencoder space,  is what we propose to enforce NES  hypotheses in practice (mainly principal and sufficient alignment), and furthermore facilitate the interpretation of the results. In principle, we may apply NES directly to the foundational model representation. However, even if NES hypotheses are satisfied in the foundational model representation space, by the emerging Linear Representation Hypothesis and weak entanglement, the principal components may not be sufficiently explicative for the experiment-specific classification, which can significantly challenge and invalidate the interpretation step, too.

---

> > ### Author Response · Authors · 2025-11-27
> >
> > We kindly ask the reviewer if their questions were answered to their satisfaction and remain available for any further discussion.

---

### Author Response · Authors · 2025-11-21
**General Answer**

We thank all the reviewers for their positive and constructive feedback, which has motivated us to immediately adjust and extend our manuscript in line with their detailed and valuable suggestions. In the updated version of the manuscript, we color-mark in red all the major modifications that are referenced in the rebuttal. We report here a concise summary:

**Experiments: extensive new method, tests, and baselines ablations**

We significantly extended the additional experiments section (Appendix E), and we restructured it into three subsections:
- Validating our assumptions given ground truth (suggested by reviewer @waNL).
- Extensive ablations on the method (suggested by reviewer @waNL). In particular:
  - varying the measurements encoder (foundational model, SAE dimension, SAE non-linearity, and training randomization),
  - varying the method hyperparameters (hypothesis testing and causal estimator),
  - varying the data-generating processes (full grid of main experiments, no effect, opposite effects, varying propensity score).
- Additional baselines (suggested by reviewer @ZiiZ).

Overall, these additional experiments extensively confirm the empirical evidence of our method’s consistency for Exploratory Causal Inference, supporting our theoretical results (unlike all other baselines).

**Theory: explicit assumption and false discovery rate convergence**

Prompted by the questions about assumption validation and the false discovery rates of our method, we clarified:
- the main theorem formulation (Appendix A.3), making the necessary assumptions more explicit, beyond vanilla linear modeling (suggested by reviewer @waNL).
- the method consistency, i.e., also FWER and FDR $\rightarrow 0$ as in standard CLT-based tests rate $O(n^{-1/2})$ (suggested by reviewer @waNL),

**Writing: general improvements**

Besides these two major changes, we polished and enriched the writing, taking into account all the proposed suggestions. Particularly:
- we added an explicit motivating problem box in the Introduction (suggested by reviewer @CiiA and @rdaU),
- we simplified the required entanglement definition (suggested by reviewer @ZiiZ)
- we added a method pseudo-code snippet (Appendix C),
- and addressed all the isolated comments individually.

---

### Meta-Review · Area_Chair_9vLc · 2025-12-23

**Summary:**

The paper tackles a setting where a randomized controlled trial is run, but the outcome of interest is not observed but unstructured measurements (e.g., images or videos) are instead available. Rather than asking “does treatment T affect outcome Y?” for a preidentified Y, the goal is to discover what latent outcomes were affected by the treatment in a data-driven way. The authors first convert raw observations into semantically meaningful representations using a pretrained foundation model, then decompose these representations into sparse, approximately interpretable representations via a sparse autoencoder. Each channel can be treated as a candidate outcome whose treatment effect can be statistically tested. The key challenge is that these neural channels are often entangled: many channels weakly respond to the same underlying causal effect. As sample size or effect strength grows, standard multiple testing (even with Bonferroni correction) will flag all entangled channels as significant, destroying interpretability. To resolve this, the paper introduces Neural Effect Search (NES), a recursive testing procedure that identifies the strongest effect first, conditions it out, and then retests the remaining channels. This enables studying one causal effect at a time, avoiding the multiple-testing issue and yielding small candidate effects that domain experts can follow up on.

However, the method still rests on strong and largely untestable representation assumptions namely the sufficiency of foundation models  of capturing the right information and monosemanticity of SAE codes. Its unclear what would happen if say a weaker foundation model was used to pull representations and how the nature of the signal changes with the quality of the inferential conclusion. While the ecological case study is now better contextualized external validation across multiple independent domains or trials would improve the claims of generality made in the paper. That said, I think these are reasonable avenues for future work.

**Reviewer Concerns:**

Some of the weaknesses identified in the second paragraph above highlight what remains in this line of work (identification, a clean mapping from capabilities of foundation models onto the ranking of inferential effects with the framework, external validitiy).

**Reviewer Scores:**

My assessment here is that the average reviewer scores would only have gone up or stayed the same based on the rebuttal.

---

### Decision · Program_Chairs · 2026-01-26

Accept (Oral)